



# Seasonal carbon dynamics of the Kolyma River tributaries, Siberia

Kirsi H. Keskitalo[1,2*], Lisa Bröder[1,3], Tommaso Tesi[4], Paul J. Mann[2], Dirk J. Jong[1], Sergio Bulte Garcia[1], Anna Davydova[5], Sergei Davydov[5], Nikita Zimov[5], Negar Haghipour[3,6], Timothy I. Eglinton[3] and Jorien E. Vonk[1*]

[1]Department of Earth Sciences, Vrije Universiteit Amsterdam, Amsterdam, The Netherlands
[2]Department of Geography and Environmental Sciences, Northumbria University, Newcastle Upon Tyne, UK
[3]Department of Earth Sciences, Swiss Federal Institute of Technology, Zürich, Switzerland
[4]National Research Council, Institute of Polar Sciences in Bologna, Italy
[5]Pacific Geographical Institute, Far East Branch, Russian Academy of Sciences, Northeast Science Station, Cherskiy, Republic of Sakha, Yakutia, Russia
[6]Laboratory of Ion Beam Physics, Swiss Federal Institute of Technology, Zürich, Switzerland

*Correspondence to*: Kirsi H. Keskitalo (kirsi.keskitalo@northumbria.ac.uk) and Jorien E. Vonk (j.e.vonk@vu.nl)

**Abstract.** Arctic warming is causing permafrost thaw and release of organic carbon (OC) to fluvial systems. Permafrost-derived OC can be transported downstream and degraded into greenhouse gases that may enhance climate warming. Susceptibility of OC to decomposition depends largely upon its source and composition which varies throughout the seasonally distinct hydrograph. Most studies to date have focused on larger Arctic rivers, yet little is known about carbon dynamics in lower order rivers/streams. Here, we characterize composition and sources of OC, focusing on less studied particulate OC (POC), in smaller waterways within the Kolyma River watershed. Additionally, we examine how watershed characteristics control carbon concentrations. In lower order systems, we find rapid initiation of primary production in response to warm weather, shown by decreasing $\delta^{13}$C-POC, in contrast to larger rivers. As Arctic warming and hydrologic changes may increase OC transfer from smaller waterways through river networks this may intensify inland water carbon outgassing.

## 1 Introduction

The Arctic is warming up to four times the rate of the global average (Meredith et al., 2019; Rantanen et al., 2022) which affects hydrology, carbon cycling and permafrost (Turetsky et al., 2019; Walvoord and Kurylyk, 2016). Terrestrial permafrost thaw adds organic carbon (OC) to fluvial systems via active layer leaching and abrupt thaw processes (e.g., river bank erosion), the former releasing predominantly dissolved OC (DOC) and the latter particulate OC (POC) (Guo et al., 2007; Schuur et al., 2015). Mineralization of terrestrially derived permafrost OC in fluvial systems adds greenhouse gases into the atmosphere enhancing climate warming (Meredith et al., 2019; Schuur et al., 2015).

Mineralization dynamics of fluvial OC are largely determined by its composition. Modern-aged DOC predominantly fuels $CO_2$ emissions from Arctic waters (Dean et al., 2020), yet permafrost DOC is preferentially degraded when present (Mann et al., 2015; Vonk et al., 2013). The fluxes, composition, and degradation of mainstem-POC have been addressed in large Arctic



rivers (e.g., Bröder et al., 2020; Guo and Macdonald, 2006; Keskitalo et al., 2022; McClelland et al., 2016), but our understanding of the carbon dynamics, especially regarding POC, and seasonality of smaller waterways are lacking.

Here, we investigate carbon characteristics (POC, DOC, dissolved inorganic carbon - DIC, stable carbon isotope $\delta^{13}$C of

35 these carbon pools, and radiocarbon $\Delta^{14}$C-POC) and water chemistry (temperature, pH, conductivity, and water isotopes $\delta^{18}$O and $\delta^2$H) in lower order streams/rivers within the Kolyma watershed (Fig. 1). We perform source-apportionment modelling to characterize sources of POC, and investigate how seasons and spatial characteristics (e.g., slope, soil OC content) affect carbon contributions in these streams. A future intensification of the Arctic hydrological cycle combined with longer growing season, earlier onset of spring freshet and on-going permafrost thaw is expected to shunt organic matter more rapidly from land into

40 lower order streams/rivers and into large river systems. It is therefore necessary to understand carbon dynamics of lower order systems in order to project future changes within Arctic rivers (Collins et al., 2021; Mann et al., 2022; Raymond et al., 2016; Stadnyk et al., 2021).

## 2 Materials and Methods

### 2.1 Study area and background

The Kolyma River drains 100 % continuous permafrost terrain (Holmes et al., 2012) with variable landscapes including wetlands, tundra and forests (Mann et al., 2012). Here, permafrost consists partially of the OC- and ice-rich Yedoma sediments, which date to the Pleistocene (Strauss et al., 2017, 2021; Zimov et al., 2006). The continental climate encompasses cold winters (January mean -32.7 °C) and mild summers (July mean 13.2 °C) (Fedorov-Davydov et al., 2018b). River hydrology is characterized by a discharge peak (>30,000 m$^3$ s$^{-1}$) during spring freshet (May–June), followed by a lower discharge (average

of 6,200 ± 3000 m$^3$ s$^{-1}$ in 2007–2017) during summer (July–August) (Shiklomanov et al., 2021). River OC concentrations follow the same pattern as discharge with higher concentrations during freshet than summer (Holmes et al., 2012; McClelland et al., 2016). All tributaries investigated in this study are partially underlain by Yedoma and located within the taiga or the tundra zone (Fig. 1) (Siewert et al., 2015; Strauss et al., 2021, 2022). Mean active layer thickness varies between catchments ranging from 154 cm in Panteleikha, 90 cm across the uplands (Y3), 65 cm at Ambolikha and 48 cm in tundra (measured at

Cape Maliy Chukochiy) (Fedorov-Davydov et al., 2018a, 2018b).



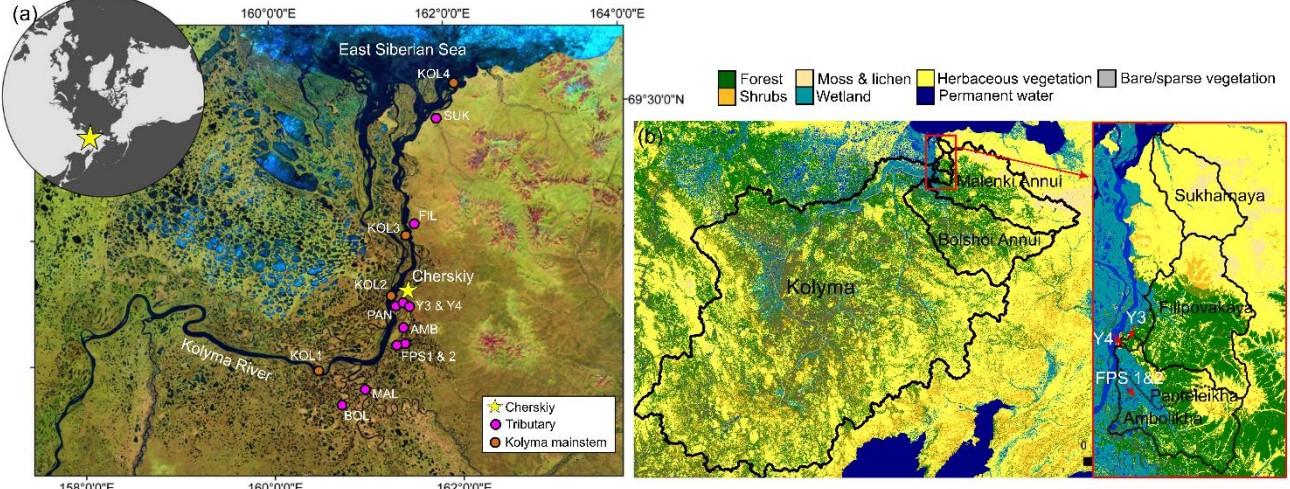

**Figure 1.** (a) Sampling locations of the Kolyma River and tributaries (i.e., lower order streams). The tributaries are Sukharnaya (SUK), Filipovkaya (FIL), Panteleikha (PAN), Malenki Annui (MAL) and Bolshoi Annui (BOL). Ambolikha (AMB), Y3 and Y4 are tributaries of Panteleikha, and floodplain streams (FPS1 and FPS2) tributaries of Ambolikha. All the sites were sampled in both seasons: summer (July-August 2018) and freshet (June 2019). Map adapted from Mann et al. (2012) (b) Land cover of the Kolyma and its tributary watersheds. Land cover classes according to Buchhorn et al. (2020).

## 2.2 Field sampling

Surface water samples were collected in summer (July–August) 2018 and spring (June) 2019 (Fig. 1, Table A1) from ~20 cm depth from the middle of the tributary river/stream (one sample per river/stream per season, total n=10 tributaries per season) and additionally in the Kolyma mainstem (n=6 in spring and n=4 in summer) using pre-rinsed 1 L Nalgene bottles, which were decanted into a 10 L sterile and pre-rinsed polyethylene bag to maximize the sample size. Water quality parameters were recorded using a multi-parameter sonde (Eijkelkamp Aquaread AP-800 in 2018, YSI Professional Plus in 2019).

Water samples were filtered (within 12 h) using pre-combusted (350 °C, 6 h) glass-fiber filters (Whatman, 0.7 µm). Prior to filtering, samples were vigorously agitated to ensure thorough particle mixing. Filters (POC samples) were frozen to -20 °C, while the filtrate (DOC samples, ~30 ml) was acidified with 30 µl of HCl (37 %) and stored cool (+5 °C). Samples for stable water isotopes ($\delta^{18}O$, $\delta^{2}H$) were filtered and stored cool (+5 °C) without headspace.

## 2.3 Analytical methods

### 2.3.1 Total suspended solids, organic carbon, and carbon isotope analyses

The amount of total suspended solids (TSS, mg L$^{-1}$) was calculated by the difference in filter weight before and after filtering, divided by the volume of water filtered. For POC concentrations, $\delta^{13}C$-POC and total particulate nitrogen (TPN) filters were freeze-dried and subsampled by punching 18 % of the 45 mm filter area and fitted into silver capsules/boats. The subsamples were treated with 1M HCl to remove inorganic carbon, and then placed into an oven at 60 °C until dry. Afterwards, the samples



were wrapped in tin capsules/boats to aid combustion during analysis. The samples were analyzed with a Thermo Fisher Elemental Analyzer (FLASH 2000 CHNS/O) coupled with a Thermo Finnigan Delta plus isotope ratio mass spectrometer (IRMS) at the National Research Council, Institute of Polar Sciences in Bologna, Italy.

For the $^{14}$C analysis, filters (see above for the subsampling method) were fumigated over 37 % HCl (72 h at 60 ℃) to remove all inorganic carbon. After fumigation, samples were neutralized of excess acid (60 ℃, a minimum of 48 h) in the presence of NaOH pellets, and subsequently wrapped in tin boats. The samples were analyzed using a coupled elemental analyzer-accelerator mass spectrometer (EA-AMS) system (vario MICRO cube, Elementar; Mini Carbon Dating System MICADAS, Ionplus, Dietikon, Switzerland) (Synal et al., 2007). The filter samples were blank corrected for constant contamination according to the method presented in Haghipour et al. (2019). The $^{14}$C analysis was carried out at the Laboratory of Ion Beam Physics at the Swiss Federal Institute of Technology (ETH), Zürich, Switzerland.

The DOC samples from summer 2018 were analyzed for OC and $\delta^{13}$C-DOC with an Aurora 1030 TOC analyzer (OI Analytical) coupled to a Delta V Advantage IRMS via a custom-built cryotrapping interface at KU Leuven, Belgium. Quantification and calibration were performed with IAEA-C6 ($\delta^{13}$C = -10.4 ‰) and an in-house sucrose standard ($\delta^{13}$C = -26.9 ‰) prepared in different concentrations. All $\delta^{13}$C data are reported in the notation relative to VPDB (Vienna Pee Dee Belemnite). The precision in duplicate samples was <5 % for DOC, and 0.2 ‰ for $\delta^{13}$C-DOC in >95 % cases. The DOC samples from freshet 2019 were analyzed for OC and $\delta^{13}$C-DOC at the North Carolina State University, Raleigh, USA. For the method details, see Osburn and St-Jean (2007).

### 2.3.2 Dissolved inorganic carbon analyses

Samples for DIC were collected by filtering 4 ml of water into pre-evacuated 12 ml exetainer (Labco, UK) containing 100 μl of $H_3PO_4$ in 2018, while in 2019, DIC samples were filtered into exetainers containing 100 μL of saturated KI and filled to the rim. The samples were stored cool (+5 ℃) and dark until analysis. Headspace $CO_2$ of the DIC samples from 2018 was analyzed using a Gasbench interfaced to a Thermo Delta V IRMS at the Northumbria University, UK. The DIC samples from 2019 were inserted into exetainers (pre-flushed with He) containing three drops of concentrated $H_3PO_4$. Subsequently, the $CO_2$ was measured with a Finnigan GasBench II interfaced with a Thermo Finnigan Delta+ mass spectrometer at the Vrije Universiteit Amsterdam, The Netherlands. Analytical standard deviation for both instruments was <0.15 ‰.

### 2.3.3 Analysis of water isotopes

We measured stable isotopes of oxygen and hydrogen ($\delta^{18}$O, $\delta^{2}$H) in water to characterize the hydrological conditions in the Kolyma River and its tributaries. Samples were analyzed with a Picarro Inc L2140-i Wavelength-scanning cavity ring-down spectrometer in replicates of seven, of which the first three were discarded to avoid carry-over effects. After a sequence of 10 samples, three in-house standards, all calibrated against international IAEA standards (VSLAP and VSMOW), were analyzed. The fourth in-house standard (KONA) was used to control precision and accuracy of the measurements (standard deviation <0.1 ‰ for $\delta^{18}$O and <2 ‰ for $\delta^{2}$H). The analysis was carried out at the Vrije Universiteit Amsterdam, The Netherlands.





### 2.4 Spatial analysis and landscape characterization

We delineated catchments using a 90 m digital elevation model (DEM) (Santoro and Strozzi, 2012) and determined mean soil OC content (SOCC) (Hugelius et al., 2013), land cover (Buchhorn et al., 2020) and calculated slope for each catchment using QGIS 3.16.1 with GRASS 7.8.4 (Fig. 1B). Prior to the spatial analysis, the DEM was pre-processed by filling all data gaps and sinks using algorithm described in Wang and Liu (2006). Two of the smallest catchments, FPS1 and FPS2, were delineated

manually using a satellite image as a template, as the DEM resolution was too coarse for delineating these small and flat catchments. For the Kolyma River watershed, we used a delineation from Shiklomanov et al. (2021). Based on size and land cover, we grouped catchments into floodplain (FPS1, FPS2), headwater (Y3, Y4), tundra (Sukharnaya, Malenki Annui), wetland (Panteleikha, Ambolikha), and forest (Bolshoi Annui, Filipovkaya) stream/rivers and Kolyma mainstem as its own.

### 2.5 Source apportionment

For the source apportionment of POC, we used a Markov Chain Monte Carlo model to quantify contributions between autochthonous (i.e., primary production), active layer, terrestial vegetation and permafrost sources. The source apportionment model accounts for uncertainties in the sources (i.e., endmembers), and estimates the residual error for the model (Stock and Semmens, 2016). We used a trophic discrimination factor (TDF) of zero assuming no discrimination (Stock and Semmens, 2016), and sampling year/season and river classes (e.g., tundra, headwater) as fixed effects for the model. The $\delta^{13}C$ and $\Delta^{14}C$

endmembers used were: autochthonous ($\delta^{13}C$ -32.6 ± 5.2 ‰, n=157; $\Delta^{14}C$ -43.2 ± 79 ‰, n=79), active layer ($\delta^{13}C$ -26.4 ± 0.8 ‰, n=56; $\Delta^{14}C$ -198 ± 148 ‰, n=60), terrestrial vegetation ($\delta^{13}C$ -27.7 ± 1.3 ‰, n=94; $\Delta^{14}C$ 97 ± 125 ‰, n=58) and permafrost ($\delta^{13}C$ -26.3 ± 0.7 ‰, n=414; $\Delta^{14}C$ -777 ± 106 ‰, n=527) according to Behnke et al. (2023), Levin et al. (2013), Vonk et al. (2012), Wild et al. (2019) and Winterfeld et al. (2015). See further details about the endmembers in Appendix A.

For the model prior, we used a Dirichlet distribution as an uninformative (on the simplex) prior. We used the model with

a chain length of 300,000, burn-in period of 200,000 and thinning of 100. The model was run in R (R Core Team, 2020) with a package *MixSiar* (Stock and Semmens, 2016). To evaluate the model convergence, we used the Gelman-Rubin and Geweke diagnostics, as well as the deviance information criteria. We report results as a mean ± standard deviation.

### 2.6 Statistical analyses

To test the difference in means in water chemistry parameters (water temperature, electrical conductivity - EC, pH and $\delta^{18}O$)

and carbon data (POC, DOC, DIC, $\delta^{13}C$-OC, $\delta^{13}C$-DIC and $\Delta^{14}C$-POC) between seasons (freshet vs summer) in the tributaries and the Kolyma mainstem, we used a Welch's t-test.

Additionally, we tested differences in above mentioned carbon parameters between differently sized streams/rivers during freshet and summer using analysis of variance (ANOVA). We grouped the rivers in small (FPS1, FPS2, Y3, Y4), midsized (Panteleikha, Ambolikha, Sukharnaya, Filipovkaya) and large rivers (Malenki Annui, Bolshoi Annui, Kolyma mainstem).



For the linear regression model of water temperature and $\delta^{13}$C-POC; $\delta^{13}$C-POC and POC-%; and $\Delta^{14}$C-POC and POC-%, we used a function *lm*. The same function was used for linear regression of spatial parameters (slope and SOCC) and OC concentrations. The significance level of all the statistical testing was 0.05. Testing was conducted in R (R Core Team, 2020). For further details on statistical methods, see Appendix A.

## 3 Results

Part of the Kolyma River mainstem data that we present here has already been reported in Keskitalo et al. (2022), including water chemistry, OC concentrations, and isotopes for organic and inorganic carbon (Tables A1, A2, A3).

### 3.1 Catchment characteristics and water chemistry

Tributary catchments ranged in size from <1 km² to nearly 60,000 km² (Table A1). Mean SOCC varied between 269 and 414 hg C/m² with the highest SOCC in the floodplain streams (FPS1, FPS2) and lowest in the tundra river Sukharnaya (Table A1).
Mean catchment slope ranged from 0.01 to 7 ° with lowest slope in the floodplain streams and highest in the tundra river Malenki Annui (Table A1). Bolshoi Annui, Filipovkaya, Y3 and Y4 were largely covered by forest (55–74 %), while Sukharnaya and Malenki Annui showed highest coverage of herbaceous vegetation (53–84 %; Fig. 1B, Table A2). The floodplain streams had the highest fraction of wetland coverage (76–80 %).

Surface water temperatures did not significantly differ between freshet and summer in the tributaries (p=0.946) or in the
Kolyma mainstem (p=0.126) but showed a larger variability during freshet (6.7 to 21 °C in tributaries; 7.2 and 18.0 °C in mainstem) than in summer (8.5 to 17 °C in tributaries; 12.5 to 15.0 °C in mainstem, Fig. 2A, Table A6). The EC and water isotope ($\delta^{18}$O) signature were lower during freshet than summer both in the tributaries (p=<0.001 and p=<0.001, respectively) and the Kolyma mainstem (p=<0.005 and p=0.048, respectively; Tables 1, A2 and A6).





**Figure 2.** (a) Surface water temperature and $\delta^{13}C$ of particulate organic carbon (POC). The linear regression for tributaries and Kolyma mainstem during both freshet and summer ($R^2$=0.33, $F_{(1,28)}$=15.07, p=<0.001; black line) and only during summer ($R^2$=0.49, $F_{(1,12)}$=13.58, p=0.003; brown line) was statistically significant while for freshet, or Kolyma mainstem and the tributaries separately, it was not. (b) The $\Delta^{14}C$-POC and $\delta^{13}C$-POC endmembers are indicated with arrows: OC from active layer (AL), terrestrial vegetation (TER), autochthonous (AU) and permafrost (PF) sources. Endmembers are according to Behnke et al. (2023), Levin et al. (2013), Vonk et al. (2012), Wild et al. (2019) and Winterfeld et al. (2015). See appendix A for more details about endmembers. (c) The $\delta^{13}C$-POC and natural logarithm (LN) of POC-% (amount of POC of total suspended solids). The linear regression for the Kolyma mainstem and tributaries (both freshet and summer, $R^2$=0.39, $F_{(1,28)}$=19.36, p=<0.001; black line) and separately for freshet was statistically significant ($R^2$=0.82, $F_{(1,14)}$=67.57, p=<0.001; blue line). Linear regression for summer only was not significant, or for tributaries and Kolyma mainstem separately. (d) The $\Delta^{14}C$-POC as a function of LN POC-%. Linear regression for summer (both Kolyma mainstem and tributaries) was significant ($R^2$=0.85, $F_{(1,12)}$=75.4, p=<0.001; brown line). Linear regression for the Kolyma mainstem or tributaries separately was not significant. All panels include data from freshet (June 2019) and summer (July–Aug 2018) in the Kolyma River mainstem and its tributaries. Part of the Kolyma data has been previously reported in Keskitalo et al. (2022).





### 3.2 Total suspended solids, carbon concentrations and isotopes of carbon

Concentrations of TSS were higher during freshet than summer at all sites (not statistically significant p=0.289; Tables 1, A6) except at FPS1, FPS2 and Y3 that showed the opposite pattern. Concentrations of POC and TPN largely followed the same trend (not statistically significant, p=0.457 and p=0.669, respectively; Table A6). In the Kolyma mainstem, POC concentrations were higher during freshet than summer (p=0.04; Table A6), while TSS and TPN showed a similar pattern (not statistically significant, p=0.071 and p=0.093, respectively). In the tributaries, DOC concentrations did not differ between

seasons (p=0.242), while DIC concentrations were lower during freshet than summer (p=<0.005; Table A6). In the Kolyma mainstem, DOC concentrations were higher during freshet than summer (p=<0.005) while DIC showed the opposite pattern (p=0.015; Table A6). Of the total carbon pool (POC, DOC and DIC), POC was the smallest carbon fraction both during freshet and summer (Fig. 3, Table A7).

During freshet, large rivers showed higher TSS and lower DOC concentrations than small ones (p=0.007 and p=0.018,

respectively), while POC, TPN and DIC did not differ between different sized rivers (Table A8). The POC-% (amount of OC of TSS) was higher in small and midsized rivers than larger ones during freshet (p=0.005) and summer (p=0.012; Fig. 4, Table A8). In summer, DOC concentrations were higher in small rivers than in large ones (p=0.003) while TSS, POC, TPN and DIC did not differ between different sized rivers (Table A8).

In the tributaries, the $\delta^{13}$C-POC did not differ between seasons (p=0.320) while $\delta^{13}$C-DOC were higher during freshet than

190 in summer (p=<0.001; Table A6). In the Kolyma mainstem, both $\delta^{13}$C-POC (p=0.01) and $\delta^{13}$C-DOC (p=0.005; Table A6) showed higher values during freshet than summer. The $\Delta^{14}$C-POC were lower (i.e., older) during freshet than summer in the tributaries (p=0.029) while in the Kolyma mainstem the trend was similar, but not statistically significant (p=0.94; Fig. 2B, Table A6). While we did not measure $\Delta^{14}$C-DOC, we report previously unpublished data (May–October 2006–2011) at FPS, Y3, Y4 and Pantheleikha (Table A10) showing that all DOC is modern. The $\delta^{13}$C-DIC was lower during freshet than summer

both in the tributaries (p=<0.001) and in the Kolyma mainstem (p=0.004; Table A6).

During freshet, small and midsized rivers showed lower $\delta^{13}$C-POC than large rivers (p=0.001) while only midsized rivers showed also lower $\delta^{13}$C-DOC (p=0.028; Table A8). During summer, the $\Delta^{14}$C-POC was higher (i.e., younger) in the small and midsized rivers than in the large ones (only significant for the small ones p=0.044; Fig. 4). In summer, there was no significant difference in $\delta^{13}$C-OC and $\delta^{13}$C-DIC between differently sized rivers (Table A8).

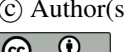

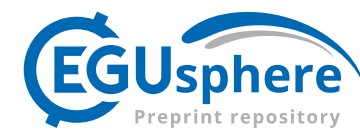

**Table 1.** Concentrations of total suspended solids (TSS), particulate and dissolved organic carbon (POC and DOC, respectively), dissolved inorganic carbon (DIC) in the tributary streams and the Kolyma River during freshet (June 2019) and summer (July–Aug 2018). Watershed types (WST) are abbreviated as headwaters (H), floodplain (FP), wetland (W), tundra (T) and forest (F). Tributary streams are abbreviated as AMB=Ambolikha, SUK=Sukharnaya, PAN=Panteleikha, FIL=Filipovkaya, MAL=Malenki Annui and BOL=Bolshoi Annui. The Kolyma mainstem is abbreviated as KOL. Also shown are stable and radioisotopes of carbon: δ¹³C of POC, DOC and DIC, and Δ¹⁴C-POC, and concentrations of total particulate nitrogen (TPN), molar ratio of POC/TPN and water isotopes (δ¹⁸O,
δH). For DIC, δ¹³C-DIC, Δ¹⁴C-POC and water isotopes (freshet n=6, summer n=4) is indicated, mean ± standard deviation between different sampling locations when applicable. Additionally, we show mean (Ave) ± standard deviation of all tributaries. Further information regarding the watersheds (e.g., size, land cover) is available in Appendix A.

| | River | WST | TSS (mg/L) | POC (μM) | POC (%) | δ¹³C-POC (‰) | Δ¹⁴C-POC (‰) | TPN (μM) | POC/TPN | DOC (μM) | δ¹³C-DOC (‰) | DIC (μM) | δ¹³C-DIC (‰) | δ¹⁸O (‰) |
|---|---|---|---|---|---|---|---|---|---|---|---|---|---|---|
| **Freshet** | Y4 | H | 7.0 | 82.1 | 14 | -33.43 | -122±21 | 8.31 | 8.5 | 887 | -26.86 | 220±9 | -16.02±0.1 | -22.81±0.1 |
| | Y3 | H | 4.7 | 26.8 | 6.9 | -29.76 | -239±24 | 2.34 | 8.7 | 1621 | -27.22 | 246±5 | -15.72±0.2 | -22.58±0.4 |
| | FPS1 | FP | 4.4 | 40.7 | 11 | -31.25 | -454±26 | 2.80 | 12.5 | 1110 | -28.56 | 376±16 | -17.67±0.3 | -23.30±0.0 |
| | FPS2 | FP | 2.8 | 65.9 | 28 | -32.92 | -268±29 | 5.51 | 10.2 | 1115 | -26.50 | 384±11 | -15.32±0.2 | -23.04±0.2 |
| | AMB | W | 9.5 | 85.4 | 11 | -32.68 | -132±26 | 10.8 | 6.6 | 994 | -29.06 | 334±9 | -18.45±0.2 | -22.75±0.0 |
| | SUK | T | 4.0 | 25.9 | 7.7 | -28.80 | -220±21 | 2.73 | 8.1 | 416 | -27.34 | 161±2 | -10.91±0.1 | -22.34±0.1 |
| | PAN | W | 13 | 77.6 | 7.4 | -33.04 | -65.1±26 | 8.04 | 8.3 | 874 | -26.86 | 207±8 | -15.93±0.5 | -22.64±0.1 |
| | FIL | F | 4.8 | 107 | 27 | -33.31 | -265±25 | 11.2 | 8.2 | 1430 | -28.47 | 282±11 | -12.54±0.0 | -22.56±0.1 |
| | MAL | T | 54 | 148 | 3.3 | -27.42 | -284±27 | 15.7 | 8.1 | 771 | -26.21 | 178±22 | -17.05±0.4 | -22.70±0.2 |
| | BOL | F | 53 | 138 | 3.1 | -27.51 | -425±299 | 12.6 | 9.4 | 770 | -26.44 | 174±4 | -16.66±0.1 | -22.88±0.1 |
| | Ave | - | 16±20 | 79.5±42 | 12±8.9 | -31.01±2.4 | -247±158 | 8.0±4.6 | 8.9±1.6 | 999±345 | -27.35±1.0 | 256±91 | -15.63±2.4 | -22.58±4 |
| | KOL | - | 39±26 | 91.6±36 | 3.6±1.9 | -27.94±1.4 | -299±71 | 8.2±2.9 | 9.4±0.9 | 708±75 | -26.70±0.4 | 283±85 | -13.09±1.6 | -22.52±2 |
| **Summer** | Y4 | H | 0.3 | 7.8 | 32 | -28.64 | -42.7±27 | 0.60 | 11.2 | 1308 | -29.51 | 535±0.3 | -15.27±0.1 | -18.61±0.1 |
| | Y3 | H | 15 | 70.5 | 5.6 | -30.65* | -177±20 | 4.96 | 12.2 | 1670 | -29.20 | n/a | n/a | -20.09±0.1 |
| | FPS1 | FP | 7.8 | 129 | 20 | -30.70 | -38.2±18 | 7.15 | 15.5 | 846 | -29.46 | 438±0.2 | -12.10±0.0 | -21.39±0.1 |
| | FPS2 | FP | 5.1 | 66.2 | 15 | -33.80 | -52.3±18 | 5.77 | 9.8 | 865 | -29.72 | 442±0.2 | -8.31 | -20.07±0.0 |
| | AMB | W | 8.1 | 85.8 | 13 | -34.00 | -63.2±17 | 9.42 | 7.8 | 806 | -29.38 | 457±0.2 | -12.38±0.0 | -21.15±0.1 |
| | SUK | T | 2.9 | 16.7 | 7.0 | -29.36 | -274±24 | 1.44 | 9.9 | 359 | -28.77 | n/a | n/a | -18.97±0.0 |
| | PAN | W | 9.2 | 143 | 19 | -34.06 | -23.9±18 | 19.0 | 6.5 | 809 | -31.16 | 313±0.1 | -11.42±0.1 | -20.95±0.0 |
| | FIL | F | 4.0 | 84.5 | 25 | -35.10 | -62.9±17 | 10.9 | 6.7 | 548 | -29.65 | 328±0.1 | -8.02±0.0 | -20.48±0.0 |
| | MAL | T | 22 | 60.0 | 3.3 | -31.25 | -348±18 | 6.29 | 8.2 | 263 | -28.91 | 281±0.1 | -9.91±0.1 | -20.20±0.1 |
| | BOL | F | 8.1 | 45.4 | 6.7 | -33.27 | -175±19 | 5.31 | 7.3 | 368 | -29.47 | 346±0.1 | -10.60±0.1 | -21.01±0.0 |
| | Ave | - | 8.2±6. | 70.9±43 | 15±9.4 | -32.08±2.2 | -126±129 | 7.1±5.2 | 9.5±2.8 | 784±441 | -29.52±0.7 | 393±88 | -11.0±2.4 | -20.29±0.9 |
| | KOL | - | 15±7 | 51.7±13 | 4.2±0.9 | -31.44±1.5 | -273±77 | 5.9±1.1 | 7.6±0.9 | 271±26 | -29.24±0.8 | 473±56 | -9.30±0.2 | -21.78±0.4 |

*average of a tributary of Y3 and Y3 mainstem upstream of the sampling site in 2018 as δ¹³C analysis at the sampling location was not successful.



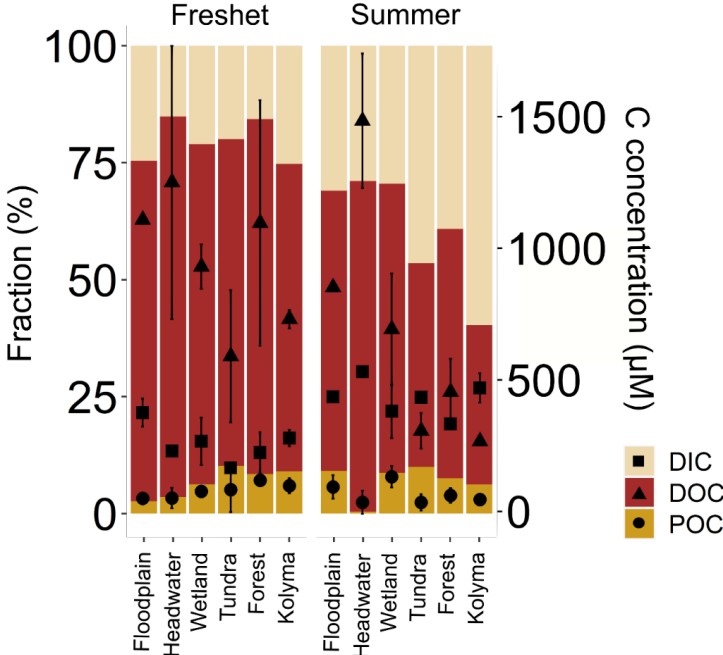

**Figure 3.** Fractions (%) of different carbon pools: particulate and dissolved organic carbon (POC and DOC, respectively) and
dissolved inorganic carbon (DIC) in the Kolyma River and its tributary rivers/streams during freshet (2019) and summer (2018).
On the right-side y-axis, concentrations of respective carbon pools are shown with square (DIC), triangle (DOC) and circle
(POC) symbols with mean ± standard deviation between samples. The tributaries are grouped based on their land cover and
size as follows (n=2 per group per season except for the Kolyma mainstem n=6 during freshet and n=4 during summer): tundra
= Sukharnaya and Malenki Annui; headwater (small, forested watersheds) = Y3, Y4; floodplain = FPS1 and FPS2; wetland
(influenced) = Ambolikha and Panteleikha; forest (larger forested watersheds) = Filipovkaya and Bolshoi Annui; Kolyma =
Kolyma mainstem. The DIC concentrations were not measured for Sukharnaya and Y3 during summer.




### 3.3 Source apportionment

Both during freshet and summer, POC was largely autochthonous in the tributaries (34–82 % and 56–92 %, respectively; Fig. 5, Table A11) and in the Kolyma mainstem (35 and 59 %, respectively). Permafrost-derived POC was higher during freshet than summer at all sites (tributaries 8–33 % during freshet and 3–22 % during summer; mainstem 34 % during freshet and 22 % during summer). Contributions from active layer and terrestrial vegetation were lowest to tributary-POC (8–24 % and 4–10% during freshet, respectively; 3–16 % and 2–7 % during summer, respectively; Fig. 5) Similarly, active layer and terrestrial vegetation contributed least to the Kolyma waters during freshet (9–22 %) and summer (6–13 %; Table A11).

### 4 Discussion

#### 4.1 Smaller tributary streams may start primary production earlier than larger rivers in the spring

In all tributaries and the Kolyma mainstem, the water isotope $\delta^{18}O$ signature significantly differed between seasons (Table A6). Lower $\delta^{18}O$ signal during freshet suggests that snowmelt was the dominant water source (Welp et al., 2005), supported by lower EC values (Table A6). However, water temperatures varied more within a season than between seasons both in the tributaries and in the Kolyma (Table A6). Air temperatures were particularly warm during freshet 2019 (see Fig. A2 for average air temperatures in 2007–2017) that was reflected as warm water temperatures especially in Filipovkaya and the floodplain streams (>20 °C). These high temperatures likely promoted a rapid onset of autochthonous production as suggested by relatively low $\delta^{13}C$-POC (up to -33.43 ‰) for the season, combined with high POC-% (11–28 %, Fig. 2C). However, in tributaries Y4, Panteleikha and Ambolikha low $\delta^{13}C$-POC occurred already prior to the high air temperatures (Table A3), suggesting that other factors such as higher nutrient fluxes during freshet likely also play a role in inducing primary production (Harrison and Cota, 1991; Holmes et al., 2012; Mann et al., 2012). Water temperature explained 33 % of the variability in $\delta^{13}C$-POC overall (higher temperature indicating lower $\delta^{13}C$-POC), while during summer it explained ~50 % of its variability (Fig. 2A), confirming that other factors affect $\delta^{13}C$-POC. Overall, freshet $\delta^{13}C$-POC was lower and POC-% higher in small and midsized rivers compared to the large ones (Fig. 4; Table A8), suggesting that river size may play a role in the timing for primary production onset during freshet. Higher input of (terrestrial) DOC (via degradation to inorganic carbon to be taken up by primary producers) and/or nutrients combined with shorter transport times may enhance primary production in smaller streams during freshet. In contrast, large rivers have longer transport times, and nutrients may already have been utilized (in headwaters), and terrestrially derived DOC degraded (Denfeld et al., 2013). Our POC data suggest that autochthonous production may start sooner in small and midsized tributaries than in large rivers during freshet.





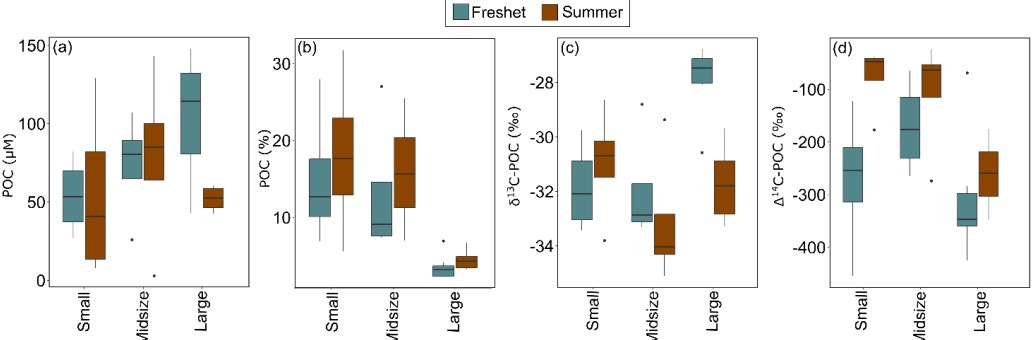

**Figure 4.** Concentrations of particulate organic carbon (POC) in (a) µM and (b) percent in small, midsized, and large rivers during freshet and summer. The (c) $\delta^{13}$C-POC and (d) $\Delta^{14}$C-POC in small, midsized, and large rivers. Boxplots show median (line), interquartile range (the box) and minimum and maximum (whiskers). For small rivers n=4 per season, for midsized rivers n=4 per season and for large rivers n=7 in summer and n=9 in freshet.

**4.2 Organic and inorganic carbon dynamics differ between the tributaries and the Kolyma River mainstem**

**4.2.1 Suspended matter dynamics**

During freshet, mean TSS and POC concentrations were higher in the large rivers than in the small tributary rivers (statistically significant only for TSS; Table A8) likely due to higher river power causing greater bank erosion (delivering sediment and POC) as well as higher turbulence promoting particle suspension (Striegl et al., 2007). Spatial characteristics such as catchment slope or SOCC did not show a linear relationship with summer-POC, indicating that other factors, such as abrupt permafrost thaw, primary production, and water temperature, likely play a more important role in driving POC concentrations (Fig. A3, Sect. 4.3). In the Kolyma, POC and $\delta^{13}$C-POC were significantly different between seasons, while in the tributaries there was no significant difference (Table A6). This likely suggests both local variability and stronger fluctuations in the tributaries that react faster to environmental changes such as high air temperatures.

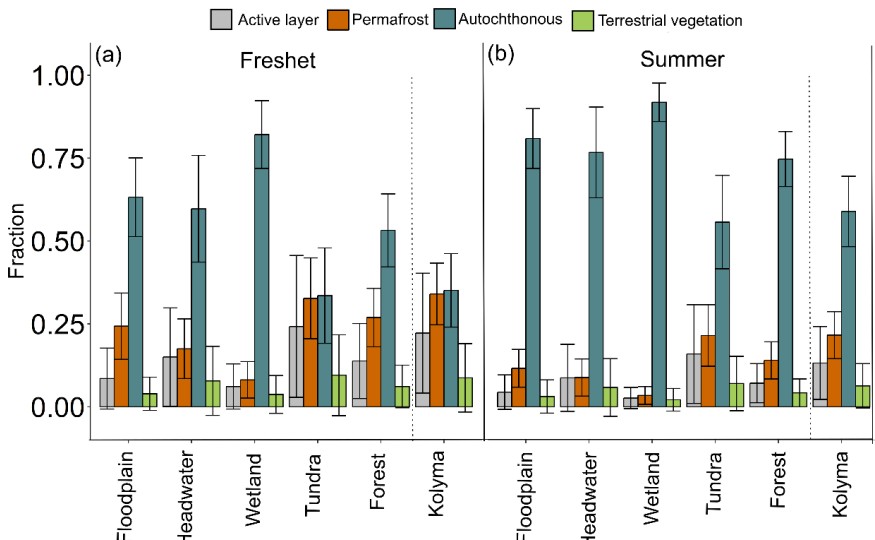

**Figure 5.** Fractions of different particulate organic carbon (POC) sources (active layer, terrestrial vegetation, autochthonous and permafrost) according to Markov Chain Monte Carlo source apportionment modelling using $\delta^{13}C$ and $\Delta^{14}C$ during (a) freshet and (b) summer. The dashed lines separate the Kolyma mainstem from the tributaries. For each catchment type (floodplain, headwater, wetland, tundra and forest) n=2 for the number of tributaries per season while for the Kolyma mainstem n=6 during freshet and n=4 during summer. The endmembers were according to Behnke et al. (2023), Levin et al. (2013), Vonk et al. (2012), Wild et al. (2019) and Winterfeld et al. (2015), see more information in the Appendix A.

**4.2.2 Dissolved matter dynamics**

Previous studies have shown that lower order streams differ from the Kolyma River in their dissolved carbon concentrations and composition (Drake et al., 2018a; Mann et al., 2012; Rogers et al., 2021). Similarly, our results show that DOC concentrations were higher in the small tributaries than in the large ones both during freshet and summer, while $\delta^{13}C$-DOC differed only between midsized and large rivers during freshet (lower for midsized rivers; Table A8). In the tributaries, SOCC predicted nearly half of the variability in DOC concentrations during summer (Fig. A3). It has been shown that the majority of DOC in the Kolyma mainstem originates from modern vegetation rather than permafrost sources (Rogers et al., 2021), potentially due to rapid degradation of permafrost-derived DOC during transit from the headwaters (Mann et al., 2015). Similarly, the $\Delta^{14}C$-DOC shows a modern signal for FPS, Y4, Y3 and Panteleikha (Table A10), implying that small stream DOC is also predominantly modern.

Both in the Kolyma mainstem and the tributaries, DIC and $\delta^{13}C$-DIC differed significantly between seasons (Table A6) and followed a previously-reported trend in fluvial systems of lower concentrations and $\delta^{13}C$-DIC during freshet than summer (Campeau et al., 2017; Waldron et al., 2007). Our Kolyma DIC concentrations were close to a previously reported concentration (Drake et al., 2018b), while $\delta^{13}C$-DIC values were ~2 ‰ higher in our study. The higher DIC concentrations during summer may reflect an increase in leaching from the active layer and/or re-mineralization of DOC, while the higher $\delta^{13}C$-DIC suggests primary production and/or partial $CO_2$ evasion, where part of the $CO_2$ is likely sourced from degraded permafrost (Campeau et al., 2017; Drake et al., 2018b; Powers et al., 2017; Waldron et al., 2007). During freshet, DIC concentrations were higher in watersheds



with higher water temperatures, a trend not observed during summer (Table 1). While higher temperatures may

increase $CO_2$ evasion and thus lower DIC concentrations (and increase $\delta^{13}$C-DIC) (Campeau et al., 2017), on-

going OC degradation potentially keeps the concentrations high. The higher $\delta^{13}$C-DIC of the Kolyma mainstem,

Sukharnaya and Filipovkaya, suggests that they may be affected by $CO_2$ evasion during turbulent freshet

conditions. At Filipovkaya, these high ratios may be partially due to primary production (i.e., biological

consumption of DIC) as the $\delta^{13}$C-POC is relatively low (Table 1). In headwater streams, contribution of OC

mineralization to the DIC pool has been suggested to be negligible relative to terrestrial input (Winterdahl et al.,

2016). Smaller streams have been shown to evade more $CO_2$ to the atmosphere than larger rivers during summer,

thus suggesting that $CO_2$ evasion from smaller streams is mainly driven by hydrological flow paths and terrestrial

OC, while in the larger rivers autochthonous production dominates as a $CO_2$ sink (Denfeld et al., 2013). Finally,

weathering, dominated by carbonates and silicates in the Kolyma watershed, may add to the DIC concentrations

(Tank et al., 2012).

### 4.3 The importance of autochthonous production: riverine POC dominates in the tributaries

Tributary-POC is mostly autochthonous both during freshet (58 ± 33 %) and summer (76 ± 27 %) indicating high

primary production, especially in summer (Fig. 5) supported by higher OC-% in small and midsized tributaries

(6.9–20 % and 5.6–32 %, respectively) than in the large rivers (~3 % and 3–7 %, respectively; Tables A8). The

$\Delta^{14}$C-POC was significantly higher (i.e., younger) in tributaries during summer than freshet, likely due to higher

primary production, while in the Kolyma $\Delta^{14}$C-POC did not significantly differ between seasons as shown

previously (Bröder et al., 2020; McClelland et al., 2016). Filipovkaya and the floodplain streams (FPS1, FPS2)

showed relatively low $\Delta^{14}$C-POC combined with high POC-% and low $\delta^{13}$C-POC (Fig. 2C–D), suggesting

incorporation of old $CO_2$ into biomass, likely originating from rapid degradation of permafrost-derived DOC

(Drake et al., 2018b). The permafrost fraction was relatively low during summer due to dominance of primary

production (Behnke et al., 2023), which was particularly pronounced in the smaller waterways (Fig. 5).

In an earlier incubation study, we showed that riverine-produced POC (with low $\delta^{13}$C-POC) in Kolyma

summer waters degrades rapidly (degradation constant k=-0.026 day$^{-1}$), while terrestrially-produced POC in

freshet waters did not show OC loss (Keskitalo et al., 2022). Furthermore, we showed that a lower initial $\delta^{13}$C-

POC corresponded to a higher POC loss. Therefore, the low $\delta^{13}$C-POC of small and midsized streams during

freshet suggests that POC may be prone to degradation, while POC degradation in the Kolyma likely lags behind

as it is still dominated by terrestrially derived POC. In smaller streams, higher water temperatures may increase

activity of bacterial communities potentially resulting in stronger degradation (Adams et al., 2010). Similarly,

leaching of terrestrial DOC and permafrost carbon may fuel stronger degradation of OC in the smaller streams

than in the larger ones (Denfeld et al., 2013).

While larger rivers may be able to emit more greenhouse gases than smaller ones given their size, smaller

rivers/streams play an important role in $CO_2$ evasion (Denfeld et al., 2013). Smaller waterways have been shown

to convey more allochthonous OC-derived $CO_2$ emissions than larger rivers (Hotchkiss et al., 2015). With the

predicted earlier onset of freshet and warmer temperatures occurring earlier in the season in the future (Meredith

et al., 2019; Stadnyk et al., 2021) (i.e., creating more favorable conditions both for primary production and OC

degradation) lower order streams could increase $CO_2$ evasion via degradation of autochthonous POC (that likely

comprises a fraction of old permafrost OC taken up by primary producers (Drake et al., 2018b), and/or enhance



degradation of allochthonous OC via priming effects (Hotchkiss et al., 2014). This may be particularly relevant in

the Arctic, where the high proportion of allochthonous permafrost OC present during freshet could be susceptible

to decomposition (Fig. 5). However, further studies are needed to decipher whether this has implications on $CO_2$

emissions in the whole system level. Furthermore, smaller rivers may transport permafrost carbon, in the form of

aquatic biomass, downstream, where its signal is mixed with modern OC sources and is not detectable anymore

(Drake et al., 2018b). Understanding dynamics of smaller rivers/streams is important given that river size may

affect their response to environmental drivers (Battin et al., 2023). On the whole, the intensification of hydrological

cycling could mean that in the future processes currently happening in lower order streams may shift towards

larger fluvial systems.

**5 Conclusions and implications**

Here, we present seasonal contrasts in water chemistry and carbon characteristics of lower order streams and the

Kolyma mainstem. However, during freshet small and midsized streams/rivers are more dynamic and seem to

respond faster to environmental changes such as air temperature increases. While POC concentrations did not

significantly differ between large and small/midsized rivers during freshet, composition of POC showed clear

differences: the $\delta^{13}$C-POC was lower and POC-% higher in small and midsized streams/rivers than in large ones,

indicating an early onset of primary production in these lower order streams. This may fuel $CO_2$ evasion via

degradation of autochthonous POC that is likely partly comprised of permafrost OC and/or prime degradation of

allochthonous OC, however, further studies are needed to discern implications on $CO_2$ emissions in a system level.

Furthermore, hydrological intensification may increase shunting and decomposition of organic matter from smaller

to larger river systems, and transport permafrost-derived OC downstream in the form of autochthonous POC. An

increased understanding of carbon and water chemistry of lower order streams and their linkages to hydrology is

therefore crucial to understand catchment-wide OC dynamics.



**Appendix A**

**Text A1. Representativeness of surface water samples**

As all our samples were of surface water, we compared our Kolyma River $\delta^{13}$C-POC data to Arctic Great Rivers Observatory (Arctic-GRO) to assess how our surface water samples would compare to depth-integrated sampling (data and sampling protocol are available in www.arcticgreatrivers.com/data, water quality) carried out since 2003 in the Kolyma River mainstem. All the water samples collected during 2003–2011 (programs PARTNERS, ARCTIC-GRO I) were depth-integrated, while samples collected between 2012 and 2021 (programs ARCTIC-GRO II-IV; data from 2020–2021 is provisional) are a combination of samples collected from the surface and at depth (sampled at depths of 4–15 m). The Arctic-GRO average ± std $\delta^{13}$C-POC for freshet (sampled in June 2004–2021) was -28.2 ± 1.4 ‰ (n=19) and for summer (sampled in July–August 2003–2021) was -29.8 ± 2.1 ‰ (n=19). In comparison, our Kolyma River mainstem $\delta^{13}$C-POC sampled during freshet (June 2019) was -27.94 ± 1.4 ‰ (n=6) and in summer (July–August 2018) was -31.44 ± 1.5 ‰ (n=4; table A2). Given that our $\delta^{13}$C-POC signature falls within the standard deviation of the depth-integrated samples we consider our samples to be sufficiently representative for the entire water column.

**Text A2. Endmembers for the source apportionment**

The endmember for autochthonous POC was according to Wild et al. (2019; $\delta^{13}$C -30.6 ± 3 ‰, n=24), Winterfeld et al. (2015; $\delta^{13}$C -30.5 ± 2.5 ‰, n=n/a), Levin et al. (2013; $\Delta^{14}$C -39.6 ± 5.5 ‰, n=73) and Behnke et al. (2023; $\delta^{13}$C -33.1 ± 4.7 ‰, $\Delta^{14}$C 106 ± 164 ‰) combined with our own POC sample collected at the Panteleikha River during an algal bloom in 2019 ($\Delta^{14}$C -26 ‰; $\delta^{13}$C -33.5 ‰, n=1). The $\delta^{13}$C endmember values from Wild et al., (2019) and Winterfeld et al., (2015) are of riverine phytoplankton from Ob and Yenisei rivers, and from Lena River, respectively, while the values from Levin et al. (2013) are of atmospheric $CO_2$ (May-August 2009-2012). Endmember values from Behnke et al. (2023) are (mostly benthic) of biofilms, algae and invertebrates from Alaska, Canada, and Svalbard. As our samples were of surface water, we combined the $\Delta^{14}$C of atmospheric $CO_2$ from Levin et al. (2013) (following the approach used by Winterfeld et al., 2015 and Wild et al., 2019) with the $\Delta^{14}$C of biofilms, algae, and invertebrates (following Behnke et al., 2023) as the autochthonous endmember. The autochthonous $\delta^{13}$C endmember was a compilation of phytoplankton (Winterfeld et al. 2015 and Wild et al. 2019) and biofilms, algae, and invertebrates (Behnke et al., 2023). For the active layer and terrestrial vegetation endmember, we used the endmembers compiled in Wild et al., (2019): endmember for active layer ($\Delta^{14}$C -197.5 ± 148.4 ‰, n=60; $\delta^{13}$C -26.4 ± 0.8 ‰, n=56) and modern vegetation ($\Delta^{14}$C 97 ± 124.8 ‰, n=58; $\delta^{13}$C -27.7 ± 1.3 ‰, n=94) The active layer and terrestrial vegetation endmembers include data from Siberia, Alaska, northern Canada, and Scandinavia. For the permafrost endmember, we combined the Ice Complex Deposit ($\Delta^{14}$C -954.8 ± 65.8 ‰, n=329) and Holocene permafrost ($\Delta^{14}$C -567.5 ± 156.7 ‰, n=138) endmember from Wild et al. (2019) with the Holocene permafrost endmember from Winterfeld et al. (2015; $\Delta^{14}$C 282 ± 133 ‰, n=60; $\delta^{13}$C -26.6 ± 1 ‰, n=40) and Vonk et al. (2012; $\delta^{13}$C -26.3 ± 0.7 ‰, n=374). All endmembers were weighed with the number of observations. We recognize that having robust endmember values is important for the best modelling results, and ideally these values would come from within or close to the studied system. While the permafrost, active layer and terrestrial vegetation endmembers are relatively well defined, scientific literature lacks well-constrained autochthonous endmembers, especially for phytoplankton. Endmembers were recently discussed in Behnke et al. (2023).



**Text A3. Statistical analyses: assumptions and hypotheses**

To test the difference in means in water chemistry parameters (water temperature, electrical conductivity - EC, pH and $\delta^{18}$O) and carbon data (POC, DOC, DIC, $\delta^{13}$C-OC, $\delta^{13}$C-DIC and $\Delta^{14}$C-POC) between seasons (i.e., freshet and summer) in the tributaries and the Kolyma River, we used a (two-sided) Welch's t-test. Our H0 hypothesis was that the means are equal between seasons and the H1 hypothesis that the means are not equal. The test significance level was 0.05. We checked the normality of the data by using the Shapiro-Wilk test and log-transformed the data in case of non-normality. For $\Delta^{14}$C-POC of tributaries, a Mann-Whitney U test was used.

To test whether there was a significant difference between small streams, midsized rivers, and large rivers regarding carbon parameters (POC, DOC, DIC, $\delta^{13}$C-OC, $\delta^{13}$C-DIC and $\Delta^{14}$C-POC), we used (one-way) analysis of variance (ANOVA) or a Kruskal-Wallis test. The floodplain streams (FPS1 and FPS2), Y3 and Y4 were classed as small streams; Panteleikha, Ambolikha, Filipovkaya and Sukarnaya as midsized rivers; and Malenki Annui, Bolshoi Annui and Kolyma mainstem as large rivers. We checked the assumptions of normality and equal variances visually and further with Shapiro-Wilk test and Breusch-Pagan test, respectively. Our H0 hypothesis was that the means are equal between different sized rivers/streams and the H1 hypothesis that the means are not all equal. With significant results, we used a Tukey's test as a *post hoc* test for ANOVA and a Dunn's test for the Kruskal-Wallis test. The significance level of all the tests was 0.05.

For the linear regression model of water temperature and $\delta^{13}$C-POC; $\delta^{13}$C-POC and POC-%; and $\Delta^{14}$C-POC and POC-%, we used a function *lm*. The same function was used for linear regression of spatial parameters (slope and soil organic carbon concentration - SOCC) and OC concentrations. The POC concentrations did not show a linear relationship with the spatial parameters, thus they were not modelled. We log transformed the DOC data prior to executing the model as well as the POC-%. For all the linear regression models, we checked the assumptions of normality and homoskedasticity of the model residuals visually and using a Shapiro-Wilk test and a Breusch-Pagan test, respectively. The significance level of the test was 0.05. All the statistical testing was executed in R (R Core Team, 2020).





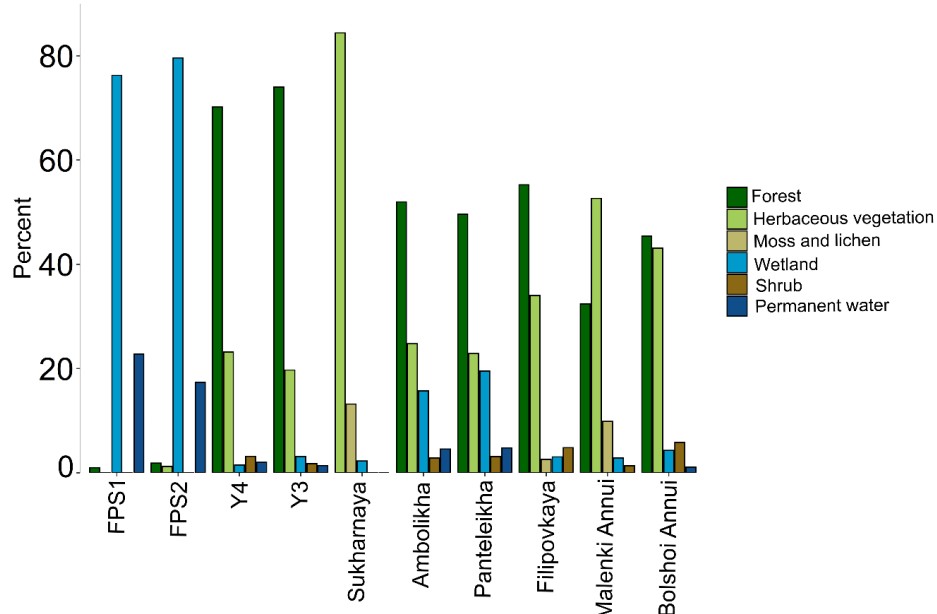

**Figure A1.** Land cover of the tributary watersheds. The watersheds are organized by their size starting from the smallest (FPS1)
on the left. The land cover types with < 1 % contribution are not included in the figure, see Table A5 for full land cover data.



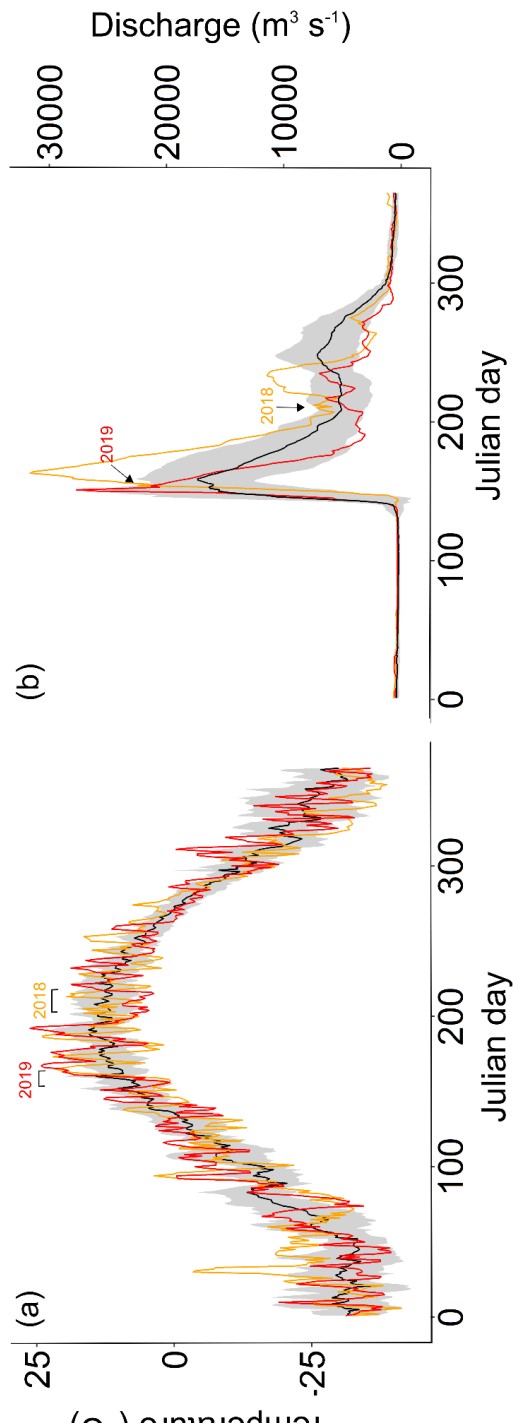

**Figure A2.** (a) average air temperature ± standard deviation (black line ± grey background) 2007–2017 in Cherskiy with air temperatures during the sampling years 2018 (orange line) and 2019 (red line). The weather data was retrieved from the Cherskiy weather station. Timing of the sampling campaigns is marked above the plot. See Table S3 for air temperatures on sampling days. (b) The average ± standard deviation of discharge measured at Kolymskoye 2007–2017 (Shiklomanov et al., 2021). Red line shows the discharge of the year 2019 and orange line the year 2018. The timing of the sampling campaigns is marked with arrows above the plot.



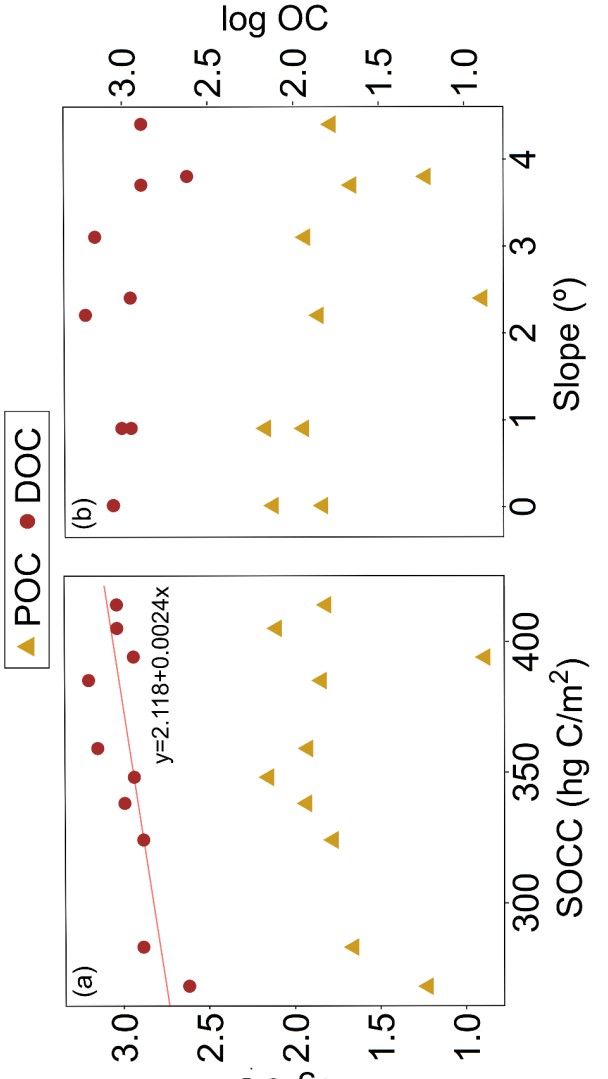

**Figure A3.** (a) Particulate and dissolved organic carbon (POC and DOC, respectively) concentration (log) and soil organic carbon content (SOCC). Linear regression for DOC was statistically significant (R²= 0.49, F(1,8)=9.59, p=0.001). (b) Concentrations (log) of POC and DOC against median slope. The regression model did not show statistically significant results. All the organic carbon data are from the Kolyma River tributaries sampled during summer 2018.



**Table A1.** Sampling coordinates and dates of the Kolyma tributaries and Kolyma mainstem during spring freshet (2019) and
summer (2018) sampling campaigns. Data from sites KOL1–KOL4 during freshet and KOL1–KOL3 during summer were
previously reported in Keskitalo et al. (2022).

| Freshet | Latitude | Longitude | Sampling date (dd/mm/yyyy) |
|---|---|---|---|
| FPS1 | N68.65100 | E161.36472 | 18/06/2019 |
| FPS2 | N68.64977 | E161.36742 | 18/06/2019 |
| Y4 | N68.74133 | E161.41393 | 08/06/2019 |
| Y3 | N68.75919 | E161.44769 | 09/06/2019 |
| Sukharnaya | N69.49534 | E161.83316 | 11/06/2019 |
| Ambolikha | N68.66421 | E161.38884 | 14/06/2019 |
| Panteleikha | N68.70052 | E161.52057 | 10/06/2019 |
| Filipovkaya | N68.92067 | E161.64552 | 16/06/2019 |
| Malenki Annui | N68.47034 | E160.83749 | 07/06/2019 |
| Bolshoi Annui | N68.46519 | E160.80356 | 07/06/2019 |
| KOL1 | N68.51782 | E160.98093 | 07/06/2019 |
| KOL2 | N68.66630 | E161.19991 | 07/06/2019 |
| KOL3 | N69.20045 | E161.44044 | 11/06/2019 |
| KOL4 | N69.62680 | E162.21594 | 11/06/2019 |
| KOL3re* | N69.20045 | E161.44044 | 16/06/2019 |
| KOL4re* | N69.62680 | E162.21594 | 16/06/2019 |
| **Summer** | | | |
| FPS1 | N68.65108 | E161.36438 | 07/08/2018 |
| FPS2 | N68.64903 | E161.36606 | 09/08/2018 |
| Y4 | N68.74216 | E161.41379 | 04/08/2018 |
| Y3 | N68.75919 | E161.44769 | 26/07/2018 |
| Sukharnaya | N69.49577 | E161.83197 | 28/07/2018 |
| Ambolikha | N68.67504 | E161.41608 | 21/07/2018 |
| Panteleikha | N68.67068 | E161.52295 | 30/07/2018 |
| Filipovkaya | N68.90665 | E161.68976 | 06/08/2018 |
| Malenki Annui | N68.45193 | E160.81279 | 01/08/2018 |
| Bolshoi Annui | N68.46015 | E160.78267 | 01/08/2018 |
| KOL1 | N68.50713 | E160.61034 | 23/07/2018 |
| KOL2 | N68.75443 | E161.27150 | 25/07/2018 |
| KOL3 | N69.20045 | E161.44044 | 28/07/2018 |
| KOL4 | N69.32058 | E161.56134 | 28/07/2018 |

*repeat measurement.





**Table A2.** Concentrations of total suspended solids (TSS), particulate and dissolved organic carbon (POC and DOC, respectively), dissolved inorganic carbon (DIC) in the Kolyma River during
freshet (June 2019) and summer (July–Aug 2018). Also shown are stable isotopes of carbon: δ¹³C of POC, DOC and DIC, and concentrations of total particulate nitrogen (TPN) and molar ratio of
POC/TPN. For Δ¹⁴C-POC, see Table A7. Mean and standard deviation between replicate samples (n=4) is shown for freshet sites KOL1–KOL4 and for summer KOL1–KOL3 (n=3, KOL3 n=4)
including analytical uncertainty for DIC and δ¹³C-DIC. For water isotopes (δ¹⁸O, δH) and summer DIC and δ¹³C-DIC only analytical error (no replicates) is shown. All data from KOL1–KOL4
during freshet and KOL1–KOL3 during summer (except DIC concentrations) were previously published in Keskitalo et al. (2022).

| Site | TSS (mg L⁻¹) | POC (μM) | POC (%) | δ¹³C-POC (‰) | TPN (μM) | POC/TPN | DOC (μM) | δ¹³C-DOC (‰) | DIC (μM) | δ¹³C-DIC (‰) | δ¹⁸O (‰) | δH (‰) |
|---|---|---|---|---|---|---|---|---|---|---|---|---|
| **Freshet** | | | | | | | | | | | | |
| KOL1 | 51±2 | 103±5 | 2.4±0.2 | -26.77±0.2 | 9.03±0.4 | 9.8±0.3 | 731±7 | -26.36±0.2 | 294±18 | -12.19±0.16 | -22.89±0.09 | -178.4±0.6 |
| KOL2 | 63±5 | 126±4 | 2.4±0.2 | -27.04±0.2 | 10.9±0.6 | 10±0.3 | 764±11 | -26.42±0.2 | 239±16 | -13.77±0.09 | -22.88±0.22 | -176.5±1.4 |
| KOL3 | 68±2 | 130±5 | 2.3±0.1 | -27.15±0.2 | 11.0±0.5 | 10±0.3 | 694±8 | -27.11±0.2 | 324±10 | -13.81±0.36 | -22.65±0.05 | -174.5±0.2 |
| KOL4 | 25±2 | 87.4±5 | 4.2±0.4 | -28.10±0.2 | 8.13±0.5 | 9.2±0.3 | 776±11 | -26.89±0.1 | 273±8 | -13.62±0.04 | -22.99±0.02 | -177.1±0.4 |
| KOL3re | 14 | 42.8 | 7.0 | -28.01 | 3.92 | 9.4 | 574 | -26.57 | n/a | n/a | -26.57±0.26 | -174.5±1.5 |
| KOL4re | 10 | 60.1 | 3.3 | -30.57 | 6.35 | 8.1 | 710 | -26.84 | 285±0.7 | -12.07±0.1 | -26.84±0.25 | -169.3±1.5 |
| **Summer** | | | | | | | | | | | | |
| KOL1 | 9.8 | 42.6±3 | 4.8 | -33.01±0.4 | 4.93±0.4 | 7.4±0.2 | 262±5 | -29.37±0.2 | 470±0.1 | -9.36±0.02 | -22.14±0.03 | -171.7±0.7 |
| KOL2 | 12±1 | 48.6±2 | 5.0±0.3 | -32.32±0.6 | 6.20±0.3 | 6.7±0.1 | 272±15 | -29.31±0.3 | 531±0.1 | -9.46±0.04 | -22.10±0.04 | -171.5±0.3 |
| KOL3 | 21±4 | 56.8±9 | 3.3±0.1 | -29.67±0.3 | 5.63±0.6 | 8.6±0.5 | 278±19 | -29.46±0.6 | 419±0.2 | -9.08±0.02 | -21.36±0.03 | -165.5±0.1 |
| KOL4 | 18 | 59.0 | 3.9 | -30.75 | 6.72 | 7.5 | 269 | -28.83 | n/a | n/a | -21.53±0.02 | -166.9±1.9 |



**Table A3.** Water chemistry parameters including water temperature (Water temp), dissolved oxygen (DO), electrical
conductivity (EC) and pH in the Kolyma River and its tributary streams/rivers during freshet (early June 2019) and summer
(July–Aug 2018). Also shown is air temperature (Air temp) on the sampling day measured at Cherskiy weather station. All
data from KOL1–KOL4 during freshet and KOL1–KOL3 during summer were previously published in Keskitalo et al. (2022).

| Freshet | Water temp (° C) | DO (mg L$^{-1}$) | EC (µM cm$^{-1}$) | pH | Air temp (° C) |
|---|---|---|---|---|---|
| FPS1 | 20.9 | 3.43 | 46.5 | 7.74 | 19.6 |
| FPS2 | 21.0 | 7.48 | 55.5 | 7.21 | 19.6 |
| Y4 | 8.8 | 10.2 | 48.4 | 8.77 | 4.9 |
| Y3 | 7.3 | 10.8 | 43.4 | 7.90 | 14.1 |
| Sukharnaya | 15.1 | 9.7 | 25.2 | 6.93 | 19.3 |
| Ambolikha | 14.9 | 7.77 | 48.3 | 7.23 | 21.2 |
| Panteleikha | 10.9 | 9.12 | 46 | 7.00 | 18.9 |
| Filipovkaya | 20.8 | 8.81 | 42 | n/a | 24.3 |
| Malenki Annui | 6.87 | 10.0 | 41.6 | 6.87 | 7.6 |
| Bolshoi Annui | 6.70 | 10.1 | 42.1 | 7.06 | 7.6 |
| KOL1 | 7.70 | 10.5 | 102.00 | 7.10 | 7.6 |
| KOL2 | 7.20 | 10.4 | 73.10 | 6.92 | 7.6 |
| KOL3 | 9.80 | 9.86 | 68.70 | 6.65 | 19.3 |
| KOL4 | 9.30 | 10.1 | 81.70 | 7.09 | 19.3 |
| KOL3re* | 13.8 | 9.39 | 104 | n/a | 24.3 |
| KOL4re* | 17.6 | 9.45 | 78 | n/a | 24.3 |
| Summer | Temp (° C) | DO (mg L$^{-1}$) | EC (µM cm$^{-1}$) | pH | Air temp (° C) |
| FPS1 | 12.8 | 3.73 | 139 | 6.61 | 4.2 |
| FPS2 | 13.3 | 9.08 | 180 | 7.26 | 10.1 |
| Y4 | 11.2 | 6.36 | 271 | 7.17 | 14.6 |
| Y3 | 12.3 | 6.29 | 211 | 6.98 | 12.8 |
| Sukharnaya | 8.5 | 9.63 | 75 | 7.77 | 7.8 |
| Ambolikha | 15.5 | 7.83 | 134 | 7.32 | 17.1 |
| Panteleikha | 14.3 | 8.32 | 139 | 6.93 | 9.2 |
| Filipovkaya | 17.0 | 10.1 | 162 | 7.47 | 7.6 |
| Malenki Annui | 14.0 | 9.41 | 185 | 7.09 | 19.1 |
| Bolshoi Annui | 13.0 | 8.95 | 169 | 7.06 | 19.1 |
| KOL1 | 15.2 | 9.25 | 255 | 7.69 | 19.4 |
| KOL2 | 15.0 | 9.43 | 249 | 7.16 | 13.2 |
| KOL3 | 13.3 | 9.00 | 222 | 7.48 | 7.8 |
| KOL4 | 12.5 | 9.16 | 228 | 7.25 | 7.8 |

*repeat samples of KOL3 and KOL4 taken on the 16th of June 2019.





**Table A4.** Watershed size, slope and soil organic carbon content (SOCC) in the top 100 cm (Hugelius et al., 2013). Slope and
SOCC are shown as mean ± standard deviation, also the slope median is shown.

| River/stream | Watershed size (km²) | Slope mean (°) | Slope median (°) | Mean SOCC (hg C/m²) |
|---|---|---|---|---|
| FPS1 | 0.33 | 0.01±0 | 0.01 | 405±10 |
| FPS2 | 0.74 | 0.01±0 | 0.01 | 414 |
| Y4 | 2.48 | 2.3±1.6 | 2.4 | 394±11 |
| Y3 | 36.09 | 2.8±3.3 | 2.2 | 385±3 |
| Sukharnaya | 956.0 | 5.7±5.6 | 3.8 | 269±124 |
| Ambolikha | 1234 | 2.6±4.9 | 0.9 | 338.3±116 |
| Panteleikha | 1782 | 2.5±4.6 | 0.9 | 355±103 |
| Filipovkaya | 1966 | 4.4±4.2 | 3.1 | 357±99 |
| Malenki Annui | 49754 | 7.0±7.4 | 4.4 | 319±103 |
| Bolshoi Annui | 56636 | 6.2±7.1 | 3.7 | 281±113 |
| Kolyma* | 657171 | 7.8±14 | 5.3 | 290±188 |

*Kolyma delineation from Shiklomanov et al. (2021).





**Table A5.** Land cover types per watershed in percentages (%). Land cover classes are according to Buchhorn et al. (2020).

| River/Stream | Forest | Wetland | Shrubs | Herbaceous vegetation | Permanent Water | Moss and lichen | Bare sparse vegetation | Urban built |
|---|---|---|---|---|---|---|---|---|
| FPS1 | 1 | 76 | 0 | 0 | 23 | 0 | 0 | 0 |
| FPS2 | 2 | 80 | 0 | 1 | 17 | 0 | 0 | 0 |
| Y4 | 70 | 1 | 3 | 23 | 2 | 0 | 0 | 0 |
| Y3 | 74 | 3 | 2 | 20 | 1 | 0 | 0 | 0 |
| Sukharnaya | 0 | 2 | <1 | 84 | <1 | 13 | 0 | 0 |
| Ambolikha | 52 | 16 | 3 | 25 | 5 | <1 | 0 | 0 |
| Panteleikha | 50 | 20 | 3 | 23 | 5 | <1 | 0 | <1 |
| Filipovkaya | 55 | 3 | 5 | 34 | <1 | 3 | 0 | 0 |
| Malenki Annui | 32 | 3 | 1 | 53 | <1 | 10 | <1 | <1 |
| Bolshoi Annui | 45 | 4 | 6 | 43 | 1 | <1 | <1 | <1 |





**Table A6.** Welch's t-test results for difference in means in electrical conductivity (EC), water temperature (Temp), pH, water isotope $\delta^{18}O$, total suspended solids (TSS), particulate and dissolved organic carbon (POC and DOC), dissolved inorganic carbon (DIC), $\delta^{13}C$ of POC, DOC and DIC and $\Delta^{14}C$-POC between seasons (freshet and summer) in the Kolyma mainstem and its tributaries. The significantly different results are highlighted in bold. The significance level was 0.05. For $\Delta^{14}C$ in tributaries, Mann-Whitney U test was used. See more details in the supplementary methods.

| Site | EC | Temp | $\delta^{18}O$ | TSS | POC | $\delta^{13}C$-POC | $\Delta^{14}C$ | TPN | DOC | $\delta^{13}C$-DOC | DIC | $\delta^{13}C$-DIC |
|---|---|---|---|---|---|---|---|---|---|---|---|---|
| Tributaries | t(9.4)= 7.36 **p=<0.001*** | t(11.6)= -0.07 p=0.946 | t(18.4)= 7.19 **p=<0.001*** | t(17.7)= -0.92 p=0.371 | t(17.7)= -0.76 p=0.457 | t(17.3)= -1.13 p=0.274 | U=79 **p=<0.029*** | t(17.7)= -0.434 p=0.669 | t(17.0)= -1.21 p=0.242 | t(15.0)= -4.6 **p=<0.001*** | t(13.6)= 3.86 **p=<0.005*** | t(15.0)= 4.28 **p=<0.001*** |
| Kolyma | t(6.2)= -15.3 **p=<0.005*** | t(6.4)= -1.8 p=0.126 | t(8.0)= -2.33 **p=0.048*** | t(5.6)= 2.22 p=0.071 | t(5.7)= 2.69 **p=0.04*** | t(6.2)= 3.7 **p=0.01*** | t(3.7)= -0.1 p=0.94 | t(6.1)= 1.989 p=0.093 | t(5.1)= 14.6 **p=<0.005*** | t(6.8)= 13.6 **p=<0.005*** | t(2.8)= -5.4 **p=0.015*** | t(3.3)= -7.4 **p=0.004*** |



**Table A7.** Fractions (%) of different carbon pools, particulate organic carbon (POC), dissolved organic carbon (DOC) and
dissolved inorganic carbon (DIC), during freshet (June 2019) and summer (July–August 2018).

| River/Stream | Freshet | | | Summer | | |
|---|---|---|---|---|---|---|
| | POC | DOC | DIC | POC | DOC | DIC |
| Floodplain | 3.66 | 76.4 | 30.2 | 7.01 | 61.4 | 31.6 |
| Headwater | 3.53 | 81.3 | 15.1 | 0.42 | 70.7 | 28.9 |
| Wetland | 6.26 | 72.7 | 21.0 | 8.76 | 61.8 | 29.5 |
| Tundra | 10.2 | 69.8 | 20.0 | 9.94 | 43.6 | 46.5 |
| Forest | 8.44 | 75.9 | 15.7 | 7.56 | 53.3 | 39.2 |
| Kolyma | 9.05 | 65.7 | 25.3 | 6.24 | 34.2 | 59.6 |





**Table A8.** Analysis of variance (ANOVA) and Kruskal-Wallis test results for difference in means in total suspended solids (TSS), particulate and dissolved organic carbon (POC and DOC), total particulate nitrogen (TPN), dissolved inorganic carbon (DIC), $\delta^{13}C$ of POC, DOC and DIC and $\Delta^{14}C$-POC between small rivers (FPS1, FPS2, Y3, Y4), midsized (mid) rivers (Panteleikha, Ambolikha, Sukharnaya, Filipovkaya) and large rivers (Malenki Annui, Bolshoi Annui and Kolyma mainstem) during freshet and summer with F statistics (from ANOVA) or H statistics (from Kruskal-Wallis test), degrees of freedom and p-values. The statistically significant (p<0.05) results are highlighted in bold. When ANOVA or Kruskal-Wallis test results were significant, post hoc test (Tukey's test for ANOVA and Dunn's test for the Kruskal-Wallis test) was conducted and their results (p-values) are listed below to indicate whether the difference was between small and midsize rivers, small and large rivers and/or midsize and large rivers. See more details in the supplementary methods.

| | TSS | POC | POC-% | $\delta^{13}C$-POC | $\Delta^{14}C$ | TPN | DOC | $\delta^{13}C$-DOC | DIC | $\delta^{13}C$-DIC |
|---|---|---|---|---|---|---|---|---|---|---|
| **Freshet** | **H(2)=** | **H(2)=** | **H(2)=** | **H(2)=** | **H(2)=** | **F(13,2)=** | **H(2)=** | **H(2)=** | **F(12,2)=** | **H(2)=** |
| Small-mid | p=1.000 | n/a | p=1.000 | p=1.00 | n/a | n/a | p=0.704 | p=1.000 | n/a | n/a |
| Small-large | **p=0.007** | n/a | **p=0.034** | **p=0.03** | n/a | n/a | **p=0.018** | p=0.369 | n/a | n/a |
| Mid-large | p=0.069 | n/a | **p=0.016** | **p=0.03** | n/a | n/a | p=0.510 | **p=0.030** | n/a | n/a |
| **Summer** | **H(2)=** | **H(2)=** | **H(2)=** | **F(11,2)=** | **H(2)=** | **H(2)=** | **H(2)=** | **H(2)=** | **F(8,2)=** | **F(8,2)=** |
| Small-mid | n/a | n/a | p=1.000 | n/a | p=1.000 | n/a | p=0.452 | n/a | n/a | n/a |
| Small-large | n/a | n/a | **p=0.044** | n/a | **p=0.044** | n/a | **p=0.003** | n/a | n/a | n/a |
| Mid-large | n/a | n/a | **p=0.034** | n/a | p=0.179 | n/a | p=0.269 | n/a | n/a | n/a |





**Table A9.** Radiocarbon measurements for particulate organic carbon (POC) including the fraction modern (Fm), $\Delta^{14}C$ and uncalibrated $^{14}C$ ages. The ETH code is a unique analysis ID assigned for each sample analyzed at the Laboratory of Ion Beam Physics, ETH, Zürich. The uncertainties are according to the method described in Haghipour et al. (2019).

| | Site | ETH code | Fm | $\Delta^{14}C$ | Age (yrs) |
|---|---|---|---|---|---|
| **Freshet** | FPS1 | 105814.1.1 | 0.55±0.01 | -454 | 4800 |
| | FPS2 | 105803.1.1 | 0.74±0.02 | -268 | 2434 |
| | Y4 | 105809.1.1 | 0.88±0.01 | -122 | 982 |
| | Y3 | 105811.1.1 | 0.77±0.01 | -239 | 2132 |
| | Sukharnaya | 105804.1.1 | 0.79±0.01 | -220 | 1927 |
| | Ambolikha | 105810.1.1 | 0.88±0.02 | -132 | 1070 |
| | Panteleikha | 105813.1.1 | 0.94±0.02 | -65 | 473 |
| | Filipovkaya | 105817.1.1 | 0.74±0.01 | -265 | 2410 |
| | Malenki Annui | 105808.1.2 | 0.72±0.01 | -284 | 2613 |
| | Bolshoi Annui | n/a | 0.58±0.17 | -291 | 2694 |
| | KOL1 | 105801.1.1 | 0.62±0.01 | -385 | 3844 |
| | KOL1 replicate 1 | 105813.1.2 | 0.68±0.01 | -321 | 3047 |
| | KOL2 | 105811.1.2 | 0.66±0.01 | -347 | 3361 |
| | KOL2 replicate 1 | 105814.1.2 | 0.67±0.01 | -332 | 3172 |
| | KOL3 | 105802.1.1 | 0.94±0.01 | -69 | 504 |
| | KOL4 | 105800.1.1 | 0.70±0.01 | -302 | 2820 |
| | KOL3re | 105815.1.1 | 0.65±0.01 | -353 | 3436 |
| | KOL4re | 105806.1.1 | 0.63±0.01 | -380 | 3774 |
| **Summer** | FPS1 | 106134.1.1 | 0.97±0.01 | -38 | 246 |
| | FPS2 | 106135.1.1 | 0.96±0.01 | -52 | 365 |
| | Y4 | 106128.1.1 | 0.97±0.02 | -43 | 285 |
| | Y3 | 102311.1.1 | 0.83±0.01 | -177 | 1499 |
| | Sukharnaya | 102304.1.1 | 0.73±0.01 | -274 | 2503 |
| | Ambolikha | 102320.1.1 | 0.94±0.01 | -63 | 458 |
| | Panteleikha | 102305.1.1 | 0.98±0.01 | -24 | 128 |
| | Filipovkaya | 102313.1.1 | 0.94±0.01 | -63 | 456 |
| | Malenki Annui | 102317.1.1 | 0.66±0.01 | -348 | 3368 |
| | Bolshoi Annui | 102318.1.1 | 0.83±0.01 | -175 | 1477 |
| | KOL1 | 104321.1.1 | 0.78±0.02 | -231 | 2040 |
| | KOL1 replicate 1 | 102314.1.1 | 0.79±0.01 | -213 | 1855 |
| | KOL1 replicate 2 | 102315.1.1 | 0.79±0.01 | -208 | 1806 |
| | KOL2 | 101944.1.1 | 0.80±0.01 | -205 | 1781 |
| | KOL2 replicate 1 | 101945.1.1 | 0.78±0.01 | -222 | 1953 |
| | KOL2 replicate 2 | 101946.1.1 | 0.77±0.01 | -239 | 2131 |
| | KOL3 | 102301.1.1 | 0.70±0.01 | -306 | 2869 |
| | KOL4 | 104322.1.1 | 0.71±0.01 | -296 | 2748 |




**Table A10.** Sampling date, concentrations of dissolved organic carbon (DOC) and Δ$^{14}$C-DOC of floodplain stream (FPS), Y4, Y3 and Panteleikha sampled during 2006–2011 (previously unpublished data; all sampling by Anya Davydova and Sergei Davydov). The location of FPS is N68.73515, E161.40408, thus different from FPS locations in this study. The ETH code is a unique analysis ID assigned for each sample analyzed at the Laboratory of Ion Beam Physics, ETH, Zürich.

| Site | Sampling date (dd/mm/yyyy) | DOC (µM) | ETH code | Δ$^{14}$C (‰) |
|---|---|---|---|---|
| FPS | 06/10/2010 | n/a | 47880.1.1 | 57.4 |
| FPS | 06/09/2011 | 613 | 48172.1.1 | 69.7 |
| FPS | 28/09/2011 | 483 | 48165.1.1 | 71.1 |
| Y4 | 05/10/2006 | 1239 | 48359.1.1 | 18.2 |
| Y4 | 15/06/2007 | 1424 | 48358.1.1 | 61.9 |
| Y4 | 31/07/2007 | 1837 | 47879.1.1 | 23.5 |
| Y4 | 07/08/2007 | 2348 | 47877.1.1 | 91.2 |
| Y4 | 16/08/2007 | 2182 | 47875.1.1 | 75.6 |
| Y4 | 25/09/2007 | 1825 | 47874.1.1 | 62.4 |
| Y4 | 10/05/2010 | n/a | 48368.1.1 | 121 |
| Y4 | 04/09/2010 | n/a | 48356.1.1 | 78.0 |
| Y4 | 11/09/2010 | n/a | 47876.1.1 | 78.7 |
| Y4 | 04/10/2010 | n/a | 47878.1.1 | 56.7 |
| Y4 | 18/08/2011 | 1358 | 48174.1.1 | 34.2 |
| Y4 | 06/09/2011 | 1015 | 48162.1.1 | 36.3 |
| Y4 | 18/09/2011 | 2116 | 48164.1.1 | 81.4 |
| Y4 | 28/09/2011 | 1517 | 48171.1.1 | 72.4 |
| Y3 | 05/10/2006 | 1544 | 48362.1.1 | 49.2 |
| Y3 | 15/06/2007 | 1550 | 48357.1.1 | 64.9 |
| Y3 | 31/07/2007 | 2220 | 47885.1.1 | 13.7 |
| Y3 | 07/08/2007 | 1691 | 47884.1.1 | 60.5 |
| Y3 | 16/08/2007 | 1717 | 47883.1.1 | 55.6 |
| Y3 | 02/10/2007 | n/a | 47886.1.1 | 96.6 |
| Y3 | 02/10/2007 | 1719 | 47881.1.1 | 80.5 |
| Y3 | 10/05/2010 | n/a | 48366.1.1 | 123 |
| Y3 | 02/09/2010 | n/a | 48367.1.1 | 87.1 |
| Y3 | 04/09/2010 | n/a | 48365.1.1 | 54.5 |
| Y3 | 18/08/2011 | 1402 | 48168.1.1 | 67.6 |
| Y3 | 05/09/2011 | 1310 | 48163.1.1 | 63.2 |
| Y3 | 11/09/2011 | n/a | 47882.1.1 | 82.6 |
| Y3 | 18/09/2011 | 1620 | 48173.1.1 | 81.5 |
| Y3 | 27/09/2011 | 1385 | 48169.1.1 | 73.6 |
| Panteleikha | 18/08/2011 | 802 | 48170.1.1 | 33.3 |
| Panteleikha | 06/09/2011 | 336 | 48360.1.1 | -5.1 |
| Panteleikha | 19/09/2011 | 546 | 48161.1.1 | 24.4 |
| Panteleikha | 28/09/2011 | 455 | 48176.1.1 | 23.2 |





**Table A11.** Source apportionment results from Markov Chain Monte Carlo analysis showing mean, standard deviation (SD) and quantiles (2.5%, 5%, 25%, 75%, 95% and 97.5%) of particulate organic carbon (POC) from active layer, permafrost, autochthonous and terrestrial vegetation (terrestrial veg) sources during freshet and summer in floodplain (FPS), headwater, wetland, tundra, forest and Kolyma mainstem. For endmembers and further details, see supplementary methods.

| | Watershed | Source | Mean | SD | 2.50% | 5% | 25% | 50% | 75% | 95% | 97.50% |
|---|---|---|---|---|---|---|---|---|---|---|---|
| **Freshet** | FPS | Active layer | 0.085 | 0.092 | 0.002 | 0.003 | 0.020 | 0.055 | 0.118 | 0.276 | 0.338 |
| | | Permafrost | 0.243 | 0.100 | 0.054 | 0.075 | 0.176 | 0.245 | 0.310 | 0.410 | 0.445 |
| | | Autochthonous | 0.632 | 0.119 | 0.386 | 0.431 | 0.556 | 0.637 | 0.712 | 0.817 | 0.853 |
| | | Terrestrial veg | 0.039 | 0.050 | 0.000 | 0.001 | 0.007 | 0.021 | 0.052 | 0.139 | 0.173 |
| | Headwater | Active layer | 0.150 | 0.148 | 0.002 | 0.004 | 0.034 | 0.104 | 0.227 | 0.468 | 0.525 |
| | | Permafrost | 0.175 | 0.090 | 0.031 | 0.044 | 0.108 | 0.168 | 0.234 | 0.332 | 0.365 |
| | | Autochthonous | 0.597 | 0.161 | 0.255 | 0.316 | 0.489 | 0.608 | 0.717 | 0.841 | 0.880 |
| | | Terrestrial veg | 0.078 | 0.104 | 0.000 | 0.001 | 0.009 | 0.034 | 0.106 | 0.305 | 0.377 |
| | Wetland | Active layer | 0.061 | 0.068 | 0.001 | 0.002 | 0.013 | 0.035 | 0.086 | 0.201 | 0.244 |
| | | Permafrost | 0.081 | 0.055 | 0.009 | 0.014 | 0.039 | 0.069 | 0.110 | 0.186 | 0.211 |
| | | Autochthonous | 0.821 | 0.102 | 0.576 | 0.625 | 0.763 | 0.839 | 0.897 | 0.955 | 0.968 |
| | | Terrestrial veg | 0.037 | 0.057 | 0.000 | 0.001 | 0.005 | 0.015 | 0.044 | 0.157 | 0.203 |
| | Tundra | Active layer | 0.327 | 0.122 | 0.076 | 0.117 | 0.241 | 0.334 | 0.413 | 0.519 | 0.555 |
| | | Permafrost | 0.335 | 0.144 | 0.092 | 0.117 | 0.225 | 0.324 | 0.436 | 0.584 | 0.634 |
| | | Autochthonous | 0.095 | 0.122 | 0.001 | 0.001 | 0.009 | 0.038 | 0.138 | 0.364 | 0.435 |
| | | Terrestrial veg | 0.026 | 0.032 | 0.001 | 0.001 | 0.006 | 0.015 | 0.034 | 0.088 | 0.116 |
| | Forest | Active layer | 0.138 | 0.113 | 0.004 | 0.007 | 0.047 | 0.112 | 0.205 | 0.359 | 0.403 |
| | | Permafrost | 0.269 | 0.088 | 0.106 | 0.126 | 0.207 | 0.267 | 0.328 | 0.415 | 0.444 |
| | | Autochthonous | 0.532 | 0.110 | 0.318 | 0.347 | 0.456 | 0.535 | 0.610 | 0.709 | 0.740 |
| | | Terrestrial veg | 0.061 | 0.064 | 0.001 | 0.002 | 0.013 | 0.038 | 0.088 | 0.189 | 0.232 |
| | Kolyma | Active layer | 0.222 | 0.181 | 0.002 | 0.004 | 0.052 | 0.195 | 0.360 | 0.544 | 0.595 |
| | | Permafrost | 0.340 | 0.093 | 0.148 | 0.179 | 0.279 | 0.346 | 0.409 | 0.478 | 0.502 |
| | | Autochthonous | 0.351 | 0.111 | 0.152 | 0.176 | 0.270 | 0.345 | 0.427 | 0.541 | 0.574 |
| | | Terrestrial veg | 0.087 | 0.103 | 0.001 | 0.001 | 0.010 | 0.041 | 0.137 | 0.313 | 0.362 |
| **Summer** | FPS | Active layer | 0.044 | 0.052 | 0.001 | 0.002 | 0.010 | 0.025 | 0.058 | 0.151 | 0.193 |
| | | Permafrost | 0.116 | 0.057 | 0.023 | 0.034 | 0.075 | 0.111 | 0.152 | 0.217 | 0.241 |
| | | Autochthonous | 0.809 | 0.090 | 0.590 | 0.650 | 0.763 | 0.823 | 0.871 | 0.926 | 0.942 |
| | | Terrestrial veg | 0.031 | 0.050 | 0.000 | 0.001 | 0.005 | 0.014 | 0.035 | 0.124 | 0.168 |
| | Headwater | Active layer | 0.087 | 0.101 | 0.001 | 0.002 | 0.015 | 0.048 | 0.119 | 0.298 | 0.378 |
| | | Permafrost | 0.088 | 0.056 | 0.011 | 0.017 | 0.046 | 0.078 | 0.119 | 0.195 | 0.228 |
| | | Autochthonous | 0.767 | 0.137 | 0.422 | 0.496 | 0.694 | 0.795 | 0.867 | 0.942 | 0.957 |
| | | Terrestrial veg | 0.058 | 0.087 | 0.001 | 0.001 | 0.007 | 0.022 | 0.067 | 0.249 | 0.329 |
| | Wetland | Active layer | 0.026 | 0.032 | 0.001 | 0.001 | 0.006 | 0.015 | 0.034 | 0.088 | 0.116 |
| | | Permafrost | 0.034 | 0.027 | 0.004 | 0.005 | 0.015 | 0.027 | 0.047 | 0.087 | 0.105 |
| | | Autochthonous | 0.918 | 0.058 | 0.759 | 0.805 | 0.895 | 0.932 | 0.959 | 0.981 | 0.987 |
| | | Terrestrial veg | 0.021 | 0.034 | 0.000 | 0.001 | 0.003 | 0.009 | 0.025 | 0.080 | 0.120 |
| | Tundra | Active layer | 0.159 | 0.149 | 0.003 | 0.006 | 0.038 | 0.114 | 0.242 | 0.456 | 0.537 |
| | | Permafrost | 0.215 | 0.093 | 0.041 | 0.064 | 0.148 | 0.213 | 0.278 | 0.371 | 0.399 |
| | | Autochthonous | 0.557 | 0.141 | 0.262 | 0.316 | 0.463 | 0.563 | 0.658 | 0.782 | 0.811 |
| | | Terrestrial veg | 0.070 | 0.082 | 0.001 | 0.002 | 0.012 | 0.040 | 0.098 | 0.246 | 0.296 |
| | Forest | Active layer | 0.071 | 0.059 | 0.004 | 0.007 | 0.029 | 0.055 | 0.099 | 0.183 | 0.222 |
| | | Permafrost | 0.140 | 0.056 | 0.051 | 0.060 | 0.099 | 0.135 | 0.174 | 0.239 | 0.262 |
| | | Autochthonous | 0.747 | 0.083 | 0.559 | 0.599 | 0.695 | 0.757 | 0.806 | 0.864 | 0.880 |
| | | Terrestrial veg | 0.042 | 0.042 | 0.001 | 0.003 | 0.012 | 0.029 | 0.057 | 0.128 | 0.159 |
| | Kolyma | Active layer | 0.132 | 0.110 | 0.003 | 0.006 | 0.043 | 0.105 | 0.191 | 0.347 | 0.405 |
| | | Permafrost | 0.216 | 0.071 | 0.077 | 0.098 | 0.166 | 0.216 | 0.264 | 0.335 | 0.357 |
| | | Autochthonous | 0.589 | 0.106 | 0.367 | 0.403 | 0.521 | 0.595 | 0.664 | 0.753 | 0.780 |
| | | Terrestrial veg | 0.063 | 0.067 | 0.001 | 0.002 | 0.013 | 0.041 | 0.091 | 0.198 | 0.244 |





**Data availability**

Data will be available within the article or in the Appendix A.

**Author contribution**

JEV and KHK lead the design of the study with contribution from LB. KHK, LB, DJJ, AD, SD and NZ conducted all the field work. KHK, LB and DJJ executed all preparatory laboratory work. NH and TIE conducted the AMS analyses, and TT and PJM analytical laboratory work regarding carbon concentrations and stable isotope analysis. KHK carried out the statistical analyses. KHK and SBG conducted the spatial analysis. KHK lead the manuscript writing with contribution from all the co-authors.

**Competing interests**

The authors declare that they have no conflict of interest.

**Acknowledgements**

We thank the staff of the Northeast Science Station (NESS) for their support during fieldwork and for providing laboratory facilities. We want to thank Karel Castro Morales and her team (Friedrich-Schiller University, Jena) and Juri Palmtag (Northumbria University) for their support in the field. Equally, we want to thank both Suzanne Verdegaal-Warmerdam and Richard Logtestijn (Vrije Universiteit Amsterdam) for their help with fieldwork preparations. Finally, we thank Niek Speetjens (Vrije Universiteit Amsterdam) for his advice with the spatial analysis. This study was funded with a starting grant from the European Research Council to Jorien E. Vonk (THAWSOME #676982), UKRI NERC to Paul J. Mann (CACOON NE/R012806/1) and NWO Rubicon to Kirsi H. Keskitalo (019.212EN.033). The study of Sergei Davydov, Anna Davydova and Nikita Zimov was partly carried out within the framework of state assignment number 122020900184-5 of the Pacific Geographical Institute of RAS.

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
