# Peer review of "Seasonal particulate organic carbon dynamics of the Kolyma River tributaries, Siberia"

_EGUsphere, 2023_

## Referee Comment (RC2)

In the submitted manuscript, 'Seasonal carbon dynamics of the Kolyma River tributaries, Siberia' by Keskitalo et al., the composition and source of organic carbon in the Kolyma River network was spatially and seasonally measured to improve the understanding of carbon dynamics in the Arctic region. The results of this study are interesting, as there is limited research on the topic in lower order Arctic streams, and important to understanding how warming and hydrological changes to the Arctic in the future may impact inland water carbon dynamics.

**General points:**

- Include information about the snow/ice melt during the freshet period in the methods. This is important in understanding the results and conclusions made.
- The results section could be restructured to separate out the spatial and seasonal aspect so that it can more clearly be followed.

**Abstract:**

L 16: What type of studies?

L20: weather → temperature

**1 Introduction:** Concise and to the point.

**2 Material and methods:**

L45-55: Could information be added in this section about the snow/ice cover in the catchment/river?

L63: Was ice/snow present during the spring sampling?

L90-95: were the methods used the same between labs?

L135-137: Report the n values for freshet and summer here.

L138-129: I assume for the ANOVA test both summer and freshet data was combined, please clarify in the text and add the n values for the three groups.

140: It could be nice to have an opening sentence as to why you used liner regression, e.g. to look at how carbon related to catchment characteristics and water chemistry.

**3 Results:** In general, this section could be restructured slightly. It was confusing to know which test (i.e. t test, Anova, linear regression) related to which result and also to separate out the spatial vs seasonal aspect. One suggestion would be separate section 3.2 out into a separate seasonal and spatial section. Further the 2.6 stats section could be rearranged to follow how the states are presented in the results, first linear regression and then the t test and anova. And in the results section when presenting the p value, you can write what test it is referring to.

L 155: larger **spatial** variability during freshet **compared to the** summer

L175 – 199: This section could be split into two, e.g. 3.2.1 Carbon across seasons and 3.2.2 Carbon across river network.

L 175: Rewrite, suggestion, "Concentrations of TSS were higher during fresher compared to summer at most sites, except at FPS1, FPS2 and Y3, but was not statistically significant (p = 0.3)"

L175 and L179 include the "not statistically significant" as part of the sentence and only the p value in ()

L227: missing "." At end of sentence

**Discussion:**

**L 260-263:** If there is snow still in the catchment during this time of year (see comments in methods section as to why this is important to mention), could the smaller streams that are more connected to the snow melt be experiencing a dilution effect?

**L296:** In section 4.1 the conclusion is that primary production starts earlier in small and warmer streams. How does this relate to trends in higher DIC in warmer waters mentioned here? Could the higher temperatures indicate more terrestrial inputs during the freshet? In particular, for the floodplain streams, which seemed to have highest DIC and temperature. Was there water pooling in the floodplain area during the freshet? Even though the streams have high primary production, they are still very hydrologically connected to the landscape. Could consider adding water to table 1 since it is referenced in the text here.

**Conclusion:** Could a sentence highlight the importance of the freshet season and how by not including it we miss an important time of year for carbon cycling.

**L 350:** Wouldn't there be an initial uptake of $CO_2$ before fueling $CO_2$ evasion?

**Figure 2:** In the legend you write which regression is significant but can this also be displayed in the graph, e.g. an * nest to the regression that is significant. In (a) the spring freshet line isn't shown and in (c) the summer line isn't shown and in (d) both freshet and summer aren't shown, write this in the legend text, e.g, "Linear regression for summer only was not significant, or for tributaries and Kolyma mainstem separately (**not shown**).". Include the n = for the freshet and summer (L1709. Can the line colors (black, brown and blue) be added to the figure legend?

**Table 1:** Ave → Avg. Consider adding water temperature to table 1 (see L 296)

**Figure 3:** Fractions → Fraction

**Figure 4:** If so, could consider adding the significance results to the figure. L 255 add "during freshet and summer" at the end of the sentence.

---

## Author Comment (AC1)

Response to Anonymous Reviewer 1 - egusphere-2023-1792. Author responses added in blue.

**Reviewer 1:**

*"The study by Keskitalo et al. aims to decipher carbon sources and their drivers in an Arctic river and its tributaries. The authors focus on particulate organic carbon (POC) and are particularly interested in differences between two seasons, spring freshet and summer, as well as differences among different sizes of streams/rivers and the mainstem of the Kolyma River in Siberia. The authors found differences in POC sources between seasons and in small tributaries compared to larger streams/rivers. With this study, the authors want to contribute to a better understanding of the carbon dynamics of the often overlooked smaller tributaries in the Arctic and their sensitivity to climatic stressors. The study provides important results for a better understanding of streams/rivers of different sizes, their carbon dynamics, and their sensitivity to climate warming. Nevertheless, I have questions about the data and the statistical analyses of the data that I would like the authors to address. Please find my comments in the order they appear in the manuscript below."*

Thank you for your review and positive response to our manuscript. We think the reviewer comments have helped to improve this manuscript and we address them one by one below.

1. **Title:** *I like that the authors formulated subheadings in the discussion that summarize the main findings. However, the title of the manuscript is very descriptive stating that the study is about "Seasonal carbon dynamics of the Kolyma River tributaries, Siberia". I would like to ask the authors to think about rephrasing the title and summarize the major result(s) and better highlight the focus of the study, the POC. Furthermore, the authors do not only report seasonal dynamics but also look at the role of the smaller sized streams compared to the mainstem of the river. A well-summarized title might also encourage more readers to look at the article.*

We agree that the title could be more detailed and suggest here a new more descriptive title for the manuscript (addition in *italics*): "Seasonal *particulate organic* carbon dynamics of the Kolyma River tributaries, Siberia".

2. **Last sentence of the abstract** *(lines 20-21): "As Arctic warming and hydrologic changes may increase OC transfer from smaller waterways through river networks this may intensify inland water carbon outgassing." I do not see the link between carbon transfer and outgassing. I miss the conversion of carbon, i.e. decomposition in the system that leads to higher $CO_2$ production through respiration, which can then cause a higher outgassing rate than at other times. I also do not believe that an earlier onset of primary production in tributaries compared to the mainstem during freshet necessarily leads to higher evasion rates. This process in itself actually leads to a reduction in $CO_2$ levels. I therefore suggest rewording the last sentence to fit the reported results or adding one or two more results from this study to make the connection here.*

We have re-written this sentence and included $CO_2$ fixation by primary producers, a process we see now that we have not highlighted enough in the manuscript. The last sentence of the abstract reads now (the additions in *italics*):

"In lower order systems, we find rapid initiation of primary production in response to warm water temperatures during spring freshet, shown by decreasing $\delta^{13}$C-POC, in contrast to larger rivers. *This results in $CO_2$ uptake by primary producers and microbial degradation of mainly autochthonous OC, however, if terrestrially-derived inorganic carbon is assimilated by primary producers, also $CO_2$ emissions may occur.* As Arctic warming and hydrologic changes may increase OC transfer from smaller waterways to *larger* river networks, *understanding carbon dynamics in smaller waterways is crucial*."

3. ***Statistical approach****: I wonder if the Welch's t-test for testing differences between the two seasons is correct here for this study design. The assumption of this test is that the two groups are independent. At the same time, the authors want to test whether spatial characteristics in the watershed influence carbon dynamics at a sampling point. This implies, in my opinion, that the authors assume that the location of the sampling point in the landscape influences water quality and carbon pools. Hence, the two samples collected in two seasons at the same site might be more similar than the others and should be paired for the statistical test. Can the authors please explain why they use the independent Welch's t-test or change their statistics if there is no justification for choosing the statistical test. I have one additional comment about the statistics. The authors use simple linear regressions to investigate how environmental factors influence carbon dynamics (Fig. 2). They state that they want to "examine how watershed characteristics control carbon concentrations." (lines 18/19). This could be done by running multiple linear regressions or linear mixed models with sampling site as a random factor (to account for their dependency) to see which factors are "most relevant" for controlling carbon dynamics. In this way, they could incorporate several independent variables.*

Thank you for these insights. Firstly, we did consider both paired and non-paired tests while establishing whether river waters sampled in spring and summer were independent and came to the conclusion to use the non-paired test. Reconsidering the test now, a paired t-test would be more appropriate as you point out that the landscape connects these sites. We have changed the Welch's test to a paired t-test (or non-parametric Wilcoxon ranked sum test if paired t-test assumptions were not met). The overall results did not change, except for $\delta^{13}$C-POC and DIC of the Kolyma mainstem (from significant to non-significant). For the sites K3 and K4 of the Kolyma River mainstem, we used an average of the two replicate samples on these sites during freshet to pair them with the same sites sampled during summer. We have updated the method, results, and discussion section accordingly along with Table A6.

Secondly, we chose to use simple linear regression (Fig. 2) instead of multiple linear regression to investigate relationships between the variables that we found interesting: these were whether water temperature can explain changes in $\delta^{13}$C-POC and how carbon isotopes ($\delta^{13}$C-POC and $\Delta^{14}$C-POC) may explain POC-%. We chose here to do simple linear regressions as we were

interested in response of both $\delta^{13}C$ to temperature and POC-% to carbon isotopes, thus having two different response variables that we were interested of.

4. The authors measured POC and particulate nitrogen (PN) and also show ratios of POC to PN in table 1. The C:N ratio can also be an indicator of algal or terrestrial material, with ratios around 8 being of algal origin and with increasing ratios being more terrestrial. Please see Figure 1 in Meyer 1994 (Meyers, Philip A. "Preservation of elemental and isotopic source identification of sedimentary organic matter." Chemical geology 114.3-4 (1994): 289-302.). Perhaps this could be included in this manuscript and highlighted in the discussions. For example in line 238.

Thank you for this comment. We agree that C/N is a good indicator of source as well and have added the following sentence to the discussion:

"While the POC pool is dominated by autochthonous OC, it is likely that allochthonous OC is also present, as suggested by POC/TPN ratios (e.g., Meyers, 1994) and our source apportionment results (see Section 4.3 and Fig. 5)."

5. At the beginning of the discussion before the first subheading (after line 229): It would be nice to insert here a summary of the main findings in relation to the main objectives formulated in the abstract (lines 17-19) and at the end of the introduction (lines 36-38).

We have included a short summary at the beginning of the discussion that reads as follows:

"*Our results show contrasting water chemistry and carbon dynamics between spring freshet and summer in the Kolyma River tributaries and mainstem. The river POC is mostly autochthonous both in the tributaries and the Kolyma mainstem during both seasons. Small and midsized rivers differ in their POC composition from large rivers with higher POC-% (freshet and summer), lower $\delta^{13}C$-POC (freshet) and higher $\Delta^{14}C$ (summer).*"

6. Lines 347-352: "While POC concentrations did not significantly differ between large and small/midsized rivers during freshet, composition of POC showed clear differences: the $\delta13C$-POC was lower and POC-% higher in small and midsized streams/rivers than in large ones, indicating an early onset of primary production in these lower order streams. This may fuel $CO_2$ evasion via degradation of autochthonous POC that is likely partly comprised of permafrost OC and/or prime degradation of allochthonous OC, however, further studies are needed to discern implications on $CO_2$ emissions in a system level." I like the conclusions the authors draw here. They highlight very nicely the most important results and implications here that I think are worth publishing. However, when primary production is higher, the authors usually also conclude that there is higher $CO_2$ evasion. I am not sure I follow this interpretation. Also in the discussion, the authors interpret their data in a similar way. Although one cannot rule out the possibility that more $CO_2$ is emitted when primary production is high, the direct consequence is that more inorganic carbon is taken up. Demars and colleagues nicely discuss the balance between primary production and respiration in streams as temperatures rise. They conclude that warming will not lead to an increase in $CO_2$ emissions in streams and rivers. See Demars et al. 2016 for a discussion on this topic (Demars, Benoît OL, et al. "Impact of warming on $CO_2$ emissions from streams

countered by aquatic photosynthesis." Nature Geoscience 9.10 (2016): 758-761.). This comment is related to the one about the last sentence of the abstract.

Thank you for this comment. We agree that while our focus was looking at potential $CO_2$ emissions from permafrost carbon degradation, we haven't highlighted inorganic carbon fixation by primary producers - an important process not to overlook. We have read the suggested paper (Demars et al., 2016) and incorporated their results to our discussion as follows:

"*While warmer water temperatures have been shown to increase microbial degradation at a similar rate as primary production, additional supply of terrestrial OC may increase degradation rates resulting in higher $CO_2$ emissions (Demars et al., 2016).*"

Additionally, we have changed the end of the abstract to highlight $CO_2$ uptake processes (see our response to question 2). Furthermore, we have highlighed $CO_2$ fixation also in the disuccion and conclusions.

---

## Author Comment (AC2)

Response to Anonymous Reviewer 2 - egusphere-2023-1792. Author responses added in blue.

**Reviewer 2:**

"In the submitted manuscript, 'Seasonal carbon dynamics of the Kolyma River tributaries, Siberia' by Keskitalo et al., the composition and source of organic carbon in the Kolyma River network was spatially and seasonally measured to improve the understanding of carbon dynamics in the Arctic region. The results of this study are interesting, as there is limited research on the topic in lower order Arctic streams, and important to understanding how warming and hydrological changes to the Arctic in the future may impact inland water carbon dynamics."

Thank you for taking time to review our manuscript. We appreciate the positive comments.

**General points:**

1. "Include information about the snow/ice melt during the freshet period in the methods. This is important in understanding the results and conclusions made".

We have included a phrase in the method section regarding snow/ice melt conditions. See also our response to questions 6 and 7.

2. The results section could be restructured to separate out the spatial and seasonal aspects so that it can more clearly be followed.

We have restructured the results section. For details, see our response to questions 12 and 14 for details.

**Abstract:**

3. *L16: "What kind of studies?*

We have specified the kind of studies we mean (in italics the change): "Most studies *on carbon dynamics* to date have focused…"

4. *L20: Weather -> temperature*

Have changed the word weather to "water temperature".

**1 Introduction**:

5. *Concise and to the point*.

Thank you.

**2 Material and methods:**

6. *L45-55: Could information be added in this section about the snow/ice cover in the catchment/river?*

We have added the following information regarding snow/ice cover in the method section:

*"During the spring freshet sampling campaign, all the rivers were ice-free during sampling. A few larger lakes in the area still had visible ice cover (5th of June 2019), but snow had largely melted and was only present in landscape depressions. The ice broke up in the Kolyma River mainstem 1st of June 2019 around the North East Science Station in Cherskiy."*

7. *L63: Was ice/snow present during the spring sampling?*

There was no ice in any of the rivers during sampling and snow had largely melted apart from occasional patches in depressions in the landscape. We have added this information to the method section, see also our response to the previous question.

8. *L90-95: were the methods used the same between labs?*

Yes, both laboratories use OI Analytical TOC analyzer connected to an IRMS (model Delta V Advantage in KU Leuven and Delta Plus$^{xp}$ in North Carolina State University) to measure DOC concentrations and $\delta^{13}$C-DOC. The method is based on wet chemical oxidation and all sample runs were accompanied with internationally renowned standards. We trust that our DOC and $\delta^{13}$C-DOC results are comparable.

9. *L135-137: Report the n values for freshet and summer here.*

The n values were added here.

10. L138-129: I assume for the ANOVA test both summer and freshet data was combined, please clarify in the text and add the n values for the three groups.

For the ANOVA test, the seasons were not combined as here we wanted to test the differences between different sized rivers separately in each season to identify differences in carbon parameters within a season rather between seasons, thus we conducted separate tests for freshet and summer. This has been clarified in the method section (as well as in Text A3) and additionally, n-values have been added.

11. 140: It could be nice to have an opening sentence as to why you used liner regression, e.g. to look at how carbon related to catchment characteristics and water chemistry.

Thank you for this suggestion, we have added the following sentence on lines 141-143:

*"We used linear regression to test how water temperature affects $\delta^{13}$C-POC, and how carbon isotopes depict POC-% to better understand river carbon dynamics. Additionally, we used linear regression to relate spatial catchment characteristics to organic carbon concentrations in rivers."*

**3  Results:**
12. *In general, this section could be restructured slightly. It was confusing to know which test (i.e. t test, Anova, linear regression) related to which result and also to separate out*

*the spatial vs seasonal aspect. One suggestion would be separate section 3.2 out into a separate seasonal and spatial section. Further the 2.6 stats section could be rearranged to follow how the states are presented in the results, first linear regression and then the t test and anova. And in the results section when presenting the p value, you can write what test it is referring to.*

Thank you for these suggestions. We have split the section 3.2 in two (see also our response to question 14 below) to make it easier to follow the results and re-arranged section 2.6 as suggested.

*13. L 155: larger **spatial** variability during freshet **compared to the** summer*

We have made the clarification to the text as suggested.

*14. L175–199: This section could be split into two, e.g. 3.2.1 Carbon across seasons and 3.2.2 Carbon across river network*

We have split this paragraph in two with the following subheadings:

"3.2.1 Seasonal carbon patterns across the catchment"

"3.2.2 Carbon patterns between different sized rivers during freshet and summer"

*15. L175: Rewrite, suggestion, "Concentrations of TSS were higher during fresher compared to summer at most sites, except at FPS1, FPS2 and Y3, but was not statistically significant (p=0.3)"*

Re-written as suggested.

*16. L175 and L179 include the "not statistically significant" as part of the sentence and only the p value in ().*

Changed accordingly.

*17. L227: missing "." At end of sentence*

Full stop added to the end of the sentence.

**Discussion:**

*18. L 260-263: If there is snow still in the catchment during this time of year (see comments in methods section as to why this is important to mention), could the smaller streams that are more connected to the snow melt be experiencing a dilution effect?*

As mentioned in response to questions 6-7, there was not a substantial amount of snow during our spring freshet sampling in 2019 thus we think that dilution was not a major reason explaining differences in TSS and POC concentrations.

19. L296: In section 4.1 the conclusion is that primary production starts earlier in small and warmer streams. How does this relate to trends in higher DIC in warmer waters mentioned here? Could the higher temperatures indicate more terrestrial inputs during the freshet? In particular, for the floodplain streams, which seemed to have highest DIC and temperature. Was there water pooling in the floodplain area during the freshet? Even though the streams have high primary production, they are still very hydrologically connected to the landscape. Could consider adding water to table 1 since it is referenced in the text here.

We agree that the smaller watersheds are hydrologically connected to the landscape and likely receive terrestrially derived DIC as shown for example in a study by Denfeld et al. (2013). It is possible that warmer water and air temperatures warm stream/river banks and thus facilitate more DIC leaching to the river. Warmer water temperatures have also been shown to increase primary production and promote faster microbial degradation. We think that all these processes are likely happening simultaneously. We have added the likely possibility of addition of terrestrial DIC to the streams as a DIC source.

As Table 1 is already rather large, we prefer not add water temperature to this table. However, we have added a reference to Table A3 (with water temperature data) here so that location of these data will be easily detectable for the reader.

**Conclusion:**

20. Could a sentence highlight the importance of the freshet season and how by not including it we miss an important time of year for carbon cycling.

We agree that this is important to highlight and have included the following sentence to the conclusions (in *italics* the addition):

"Here, we present seasonal contrasts, *including the hydrologically important spring freshet period*, in water chemistry and carbon characteristics of lower order streams and the Kolyma mainstem."

21. L350: Wouldn't there be an initial uptake of CO2 before fueling CO2 evasion?

That is correct and we agree that it is important to include, thus we have added mention of $CO_2$ uptake processes. See also our response to comment #6 of Reviewer 1.

22. Figure 2: In the legend you write which regression is significant but can this also be displayed in the graph, e.g. an * nest to the regression that is significant. In (a) the spring freshet line isn't shown and in (c) the summer line isn't shown and in (d) both freshet and summer aren't shown, write this in the legend text, e.g, "Linear regression for summer only was not significant, or for tributaries and Kolyma mainstem separately (not shown).". Include the n = for the freshet and summer (L1709. Can the line colors (black, brown and blue) be added to the figure legend?

We have added an asterisk and p-values to the statistically significant regression lines in all panels and clarified in the text that the non-statistically significant regression lines are not shown. We have also added line colors to the legend.

23. Table 1: Ave -> Avg. Consider adding water temperature to table 1 (see L 296)

We have changed the abbreviation Ave to Avg.

24. Figure 3: Fractions -> Fraction

Changed.

25. Figure 4: If so, could consider adding the significance results to the figure. L 255 add "during freshet and summer" at the end of the sentence.

We have added 'during freshet and summer' to the end of the sentence. We decided not to include any indication about significant results to the figure itself as we thought that it would be difficult and potentially confusing to show in one panel differences between three groups (small and midsized, midsized and large and small and large). However, we have added this information to the caption of the figure.